# Niche availability and competitive loss by facilitation control proliferation of bacterial strains intended for soil microbiome interventions

Senka Čaušević [1], Manupriyam Dubey [1], Marian Morales [1], Guillem Salazar [2], Vladimir Sentchilo [1], Nicolas Carraro [1], Hans-Joachim Ruscheweyh [2], Shinichi Sunagawa [2] & Jan Roelof van der Meer [1] ✉

Microbiome engineering – the targeted manipulation of microbial communities – is considered a promising strategy to restore ecosystems, but experimental support and mechanistic understanding are required. Here, we show that bacterial inoculants for soil microbiome engineering may fail to establish because they inadvertently facilitate growth of native resident microbiomes. By generating soil microcosms in presence or absence of standardized soil resident communities, we show how different nutrient availabilities limit outgrowth of focal bacterial inoculants (three Pseudomonads), and how this might be improved by adding an artificial, inoculant-selective nutrient niche. Through random paired interaction assays in agarose microbeads, we demonstrate that, in addition to direct competition, inoculants lose competitiveness by facilitating growth of resident soil bacteria. Metatranscriptomics experiments with toluene as selective nutrient niche for the inoculant *Pseudomonas veronii* indicate that this facilitation is due to loss and uptake of excreted metabolites by resident taxa. Generation of selective nutrient niches for inoculants may help to favor their proliferation for the duration of their intended action while limiting their competitive loss.

Microbiomes, the collective composite of microbial taxa and their habitats, play crucial roles in the functioning and health of their hosts or environments. Imbalanced or dysfunctional microbiomes pose a great challenge as they may present unstable developmental trajectories with a greater tendency for outgrowth of pathogens, reduced diversity, and/or diminished key ecological processes[1-4]. Consequently, there is an important need to understand whether and how interventions can be directed to equilibrate a microbiome's compositional or functional trajectory[5,6].

A classical intervention to alter microbiome composition is the inoculation of one or more microbial strains with specific functionalities[7], for example, to provide pollutant-degrading capacity for a contaminated site[8,9] or to enhance secondary metabolite production to deter potential plant pathogens[10-12]. However, despite the conceptual simplicity, such inoculations mostly fail to produce the intended effect[7,12,13]. The reasons for failure can be manifold but are reflected in the poor proliferation of the inoculated strain(s) within the targeted microbiome. Typically, probiotic therapies compensate

[1]Department of Fundamental Microbiology, University of Lausanne, 1015 Lausanne, Switzerland. [2]Department of Biology Institute of Microbiology, ETH Zurich, Vladimir-Prelog-Weg 4, 8093 Zurich, Switzerland. ✉e-mail: Janroelof.vandermeer@unil.ch

for this effect by frequent (e.g., daily) reapplication of the strain mixture to temporally manifest and maintain the required function[14,15]. Nonetheless, the fundamental questions of why newly inoculated strains often struggle to establish in target microbiomes and, accordingly, why taxa-complexity provides microbiomes with invasion resistance remain largely unresolved[16].

Niche availability is thought to be an important factor determining successful inoculant proliferation[5,17,18]. The growth and development of a species-diverse microbiota is likely to exploit all carbon, nutrient, and spatial niches in their habitat leaving few open niches for incoming species to proliferate[18–20]. As a result of emerging functionalities by the developing microbiome the habitat conditions may change further, for example, via niche pre-emption[21], disfavoring easy access and opportunities for new strains to grow. Furthermore, cells of freshly inoculated strains may not find appropriate spatial niches to be protected against predation, e.g., by protists[22], resulting in their general decline[23], or fail to establish profitable interactions with resident microbiota species[7]. Many of these arguments have not been subjected to systematic experimental testing of both the receiving microbiome and the introduced inoculant strain, and mechanistic understanding of inoculant proliferation is based on a small number of studies with, frequently, invasive pathogenic strains[24–26]. We ourselves and others have recently argued how selective inoculation studies into defined microbiota, a concept we named N + 1/N − 1 engineering[16], can be used to uncover underlying mechanisms of community assembly and development to guide future intervention practices.

The major objectives here were to study the importance of potential (nutrient and spatial) niche availability and interspecific interactions on the proliferation of soil inoculants intended for use either to reinforce xenobiotic compound metabolism or to provide plant-growth beneficial functions. Four inoculants were selected: two are capable of degrading monoaromatic compounds such as toluene (*Pseudomonas veronii* 1YdBTEX2 and *Pseudomonas putida* F1)[27,28], one is a plant-beneficial bacterium (*Pseudomonas protegens* CHA0)[29], and one was selected as a non-soil strain control (*Escherichia coli*). To test the effects of niche availability we cultured standardized, taxonomically diverse, naturally-derived soil communities (NatComs) inside sterile soil microcosms according to a previously developed protocol[30]. The microcosms contain sterilized silt matrix from a river bank sediment augmented with a complex organic compound fraction extracted directly from soil to provide nutrients for bacterial growth, whereas the soil pores provide spatial niches for the developing communities and/or inoculants[30]. By separate or concomitant inoculation of NatComs and inoculants, we aimed to generate three nutrient niche conditions. First, in absence of NatComs, all potential niches were assumed to be available for the inoculant (high niche availability), serving as a control for the inoculant's capacity to proliferate in the soil habitat. Second (intermediate niche availability), by simultaneous introduction into the soil microcosms, inoculants would be in direct competition with NatCom taxa for nutrient and spatial niches. Third (low niche availability), where the majority of available nutrient niches were assumed to have been depleted following precolonization by NatCom, after which the inoculant was introduced, and, potentially, even spatial niches would be fewer. Then in addition, we tested the effect of generating a selective nutrient niche for *P. veronii* in the soil microcosms in the form of bioavailable toluene, which we hypothesized could give the inoculant a growth advantage despite the competition for the inherent soil nutrient and spatial niches. Inoculant and NatCom populations were followed over time in their soil habitats, to estimate the realized niche from the extent of inoculant proliferation, and quantify any resulting changes in community diversity. To better understand the potential impact of biotic interactions on inoculant proliferation, we studied randomized paired-growth interactions between inoculant and soil bacteria in micro-agarose beads[31] on toluene and different defined as well as undefined substrates (i.e., soil

organic fraction extract). Finally, by metatranscriptomic analysis of enriched expressed gene functions, we evaluated in situ metabolic interactions by resident bacteria in a broader variety of soils as a consequence of *P. veronii* growth and its metabolism of toluene. Our results clearly show a generally poor proliferation of soil inoculants in presence of resident communities as a result of limited nutrient niche availability and competition. We also find that soil inoculants have a tendency to further lose productivity as a result of metabolite sharing with resident soil bacterial taxa (which we call here *competitive loss by facilitation*). However, the provision of an inoculant specific nutrient niche improves its survival and allows its functional integration into the resident microbiome network.

## Results

### Producing taxa-diverse soil-cultured microbial communities in growing or stable states

To investigate the role of nutrient niche availability on the potential for inoculants to proliferate in a soil habitat within a taxa-diverse resident soil microbiota, we generated soil microcosms either operated axenically, or colonized with two standardized types of community physiological states: (i) a growing resident community (GROWING) and (ii) a steady-state resident community (STABLE). Our hypothesis was that the introduction of an inoculant simultaneously with a resident community (GROWING) into a niche-replete soil habitat would give all strains equal opportunity to utilize available nutrient niches promoting proliferation, whereas in the case of a STABLE resident community inoculants would find fewer available niches, which would restrict their proliferation. Proliferation in the microcosms under axenic conditions in absence of the resident community served as a control for the maximum accessible nutrient niche for each of the inoculants. We first verified whether compositions of GROWING and STABLE resident soil microbiota were similar, in order to reasonably ensure that differences in inoculant proliferation would be due to nutrient niche availability rather than taxa compositional differences.

Resident soil microbiota was grown from existing soil communities (NatComs)[30], which had been maintained for 1.5 years in soil microcosms. The soil microbiota were revived by mixing the colonized soil 1:10 (*v/v*) into freshly prepared soil microcosms (Fig. 1a). In total 50 similar-sized microcosms were produced (to avoid any upscaling effects), of which 5 were used for community analysis in Phase 1 and the rest for Phase 2 incubations; see below. Microcosms all consisted of a sterilized silt matrix supplemented with a freshly sterilized aqueous extract of forest top-soil organic matter (Fig. 1a, *Methods*). Diluted NatComs were incubated for one month during which rapid growth was observed in the first days post dilution followed by a stabilization of the community size at $8 \times 10^8$ cells g$^{-1}$ soil (Fig. 1b). This density is comparable to the previously observed NatCom community size[30] and is similar to typical microbial cell densities in top soils[32,33], suggesting complete utilization of the available nutrient niches under the operated habitat conditions (i.e., aerobic, 10 % *w/v* moisture). Community succession was characterized by an initial increase in the most abundant phyla Firmicutes and Proteobacteria followed by slower-growing taxa, such as those from the Planctomycetes phyla (Fig. 1c). Lesser abundant members of Verrucomicrobia, Bacteroidetes, and Actinobacteria were also detected in the revived NatComs after one month (Fig. 1c). NatCom succession led to a temporary decrease in detectable Chao1 richness and Shannon diversity indices, which slowly increased and stabilized (Fig. 1b, Supplementary Fig. 1). After one month, the revived NatComs again resembled their starting material (Fig. 1b, Chao1 and Shannon indices, $P = 0.9840$ and $P = 1$; Supplementary Data 1), and Principal Coordinates Analysis based on Unifrac distances at species level showed that the revived NatCom community went through a development phase, more or less returning at Day 23 to the ordination points of Day 0 (Fig. 1d, $P = 0.001$ and $r^2 = 0.815$ for the time/ group effect; $P = 0.054$ in *post hoc* pairwise PERMANOVA comparison

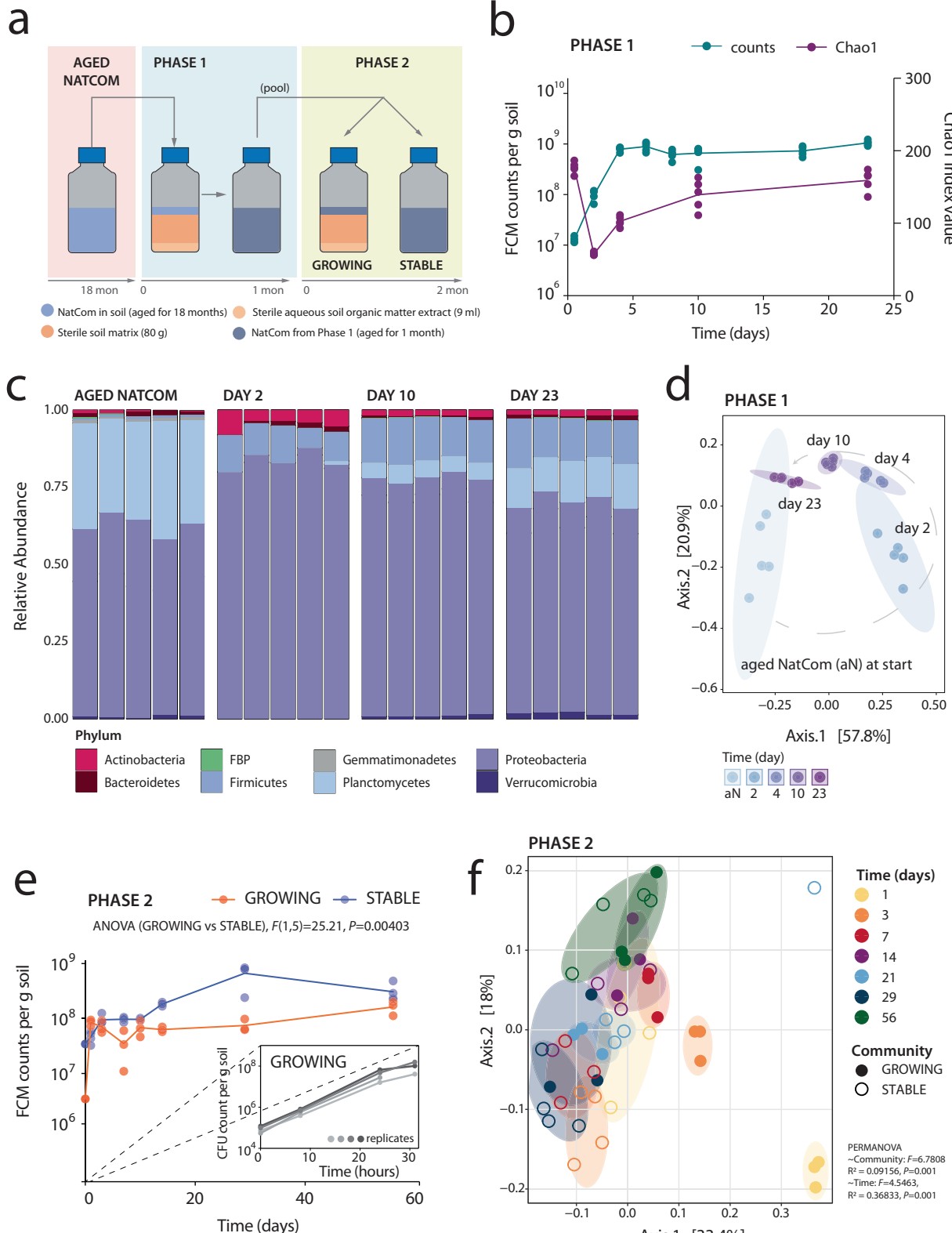

of Day 23 to Aged NatCom, Supplementary Data 1). *Inter alia*, this experiment also showed that cultured taxonomically-diverse soil communities can be maintained within the soil for extended time periods without extensive taxa loss.

At the end of Phase 1 (Fig. 1a), all soil microcosms were pooled (the remaining 45 bottles, see above) and divided into two sets that served as resident background for testing inoculant proliferation. This ensured equal starting material in Phase 2 for all microcosms. One part of the pooled material was again diluted (1:10 *v/v*) in fresh soil matrix with aqueous soil organic matter extract to create a GROWING condition in Phase 2 (Fig. 1a), whereas the STABLE condition was produced purely from pooled and mixed material from the end of Phase 1 filled in new bottles without any new supplements (Fig. 1a). The GROWING NatCom showed rapid growth in the first two days (Fig. 1e, inset) to a

**Fig. 1 | Producing standardized soil microbial communities for testing inoculant niche availability. a** Experimental approach to obtain resident soil microbial communities (NatComs) in either a growing or stable state. In Phase 1, 1.5 year old, stored soil NatComs were revived by diluting material into fresh sterile soil microcosms and incubating for one month. After one month, pooled, revived NatComs were used for Phase 2, either directly (community condition STABLE) or diluted 1:10 ($v/v$) with fresh microcosm material (condition GROWING). **b** Community growth flow cytometry (FCM) counts (left y-axis, cyan) and Chao1 index values for richness at amplicon sequence variant (ASV) level during Phase 1 (right y-axis, purple line). Dots show individual replicate measurements. Replicates are the same for all time points (repeated measures) except for $T_0$, which is a sample of the total pool used for Phase 1 inoculation. **c** Phyla relative abundance changes during Phase 1. Individual stacked bars per time point show biological replicate compositions. The most abundant phyla are colored differently, as specified in the color key. **d** Trend in community development during Phase 1 (from light blue to dark magenta along the gray dashed line) represented on principal coordinate analysis (PCoA) ordination of Unifrac distances (ASV level). Ellipses group replicates at each time point with 95% confidence level (multivariate t-distribution). **e** Mean Phase 2 community cell densities over time (FCM counts; dots indicating individual biological replicates). *P-* and *F-*values refer to cell density differences between GROWING and STABLE NatCom sizes (two-way repeated measures ranked ANOVA, Supplementary Data 1). Inset plot shows repeated experiment for community growth during the first 30 h upon dilution 1:100 ($v/v$) with fresh microcosm material quantified using colony forming units (CFU) per g of soil (shades of gray represent biological replicates). **f** PCoA ordination of Unifrac distances (species level) of Phase 2 GROWING and STABLE NatComs, colored by sampling time points. Note the conversion of both communities at later time points. Ellipses group replicates with 95% confidence level (colors correspond to figure legend). PERMANOVA tests for incubation time and community effects.

final average 47.2-fold size increase after 56 days (Fig. 1e). Cell densities in the STABLE NatComs also increased, perhaps because of the pooling and mixing process at the end of Phase 1, but less than in the GROWING NatCom (2.8-fold after 10 days and 8.9-fold at day 56; Fig. 1e). During Phase 2, the GROWING NatCom cell densities remained on average two-fold lower than in the STABLE NatComs (Fig. 1e, $P = 0.00403$, $F(1,5) = 25.21$, two-way repeated measures ranked ANOVA). Given the NatCom growth in Phase 1, this difference to Phase 2 was unexpected, but which may be due to the influence of slow-growing taxa. As expected for their different states, GROWING and STABLE NatComs differed in Chao1 richness and Shannon during the first week, but became more similar from Day 14 onwards (Supplementary Fig. 1). This apparent lower richness is due to the saturation of sample sequencing depth by faster-growing species and a subsequent reduction in the detectable species count. Over time slower-growing species become more abundant and are redetected. In ordination analysis, both GROWING and STABLE NatComs converged over time (Fig. 1f), indicating that they became compositionally similar. We thus concluded that, while the dynamic succession of GROWING and STABLE NatComs varied and their evenness remained distinct, their richness and compositional properties remained similar. This meant we could use both community states for testing the effects of nutrient niche availability on inoculant proliferation, without being confounded by compositional dissimilarities.

## Inoculant establishment is dependent on niche availability and inoculant characteristics

Next, we tested the four selected bacterial inoculants (*P. putida*, *P. protegens*, *P. veronii*, and *E. coli*) for their potential to proliferate in the soil microcosms under the three different nutrient niche availabilities: high, axenic growth of the inoculant alone (Fig. 2a, ALONE); intermediate, inoculant grows concomitantly with the freshly diluted NatCom and competes for nutrient niches (Phase 2 GROWING, starting at $t = 0$ as in Fig. 1e); and low, inoculant is introduced only after NatCom establishment and most nutrient niches are depleted (STABLE, also starting at $t = 0$ in Fig. 1e). Axenically, all three pseudomonads reached similar cell densities ($1-3 \times 10^7$ cells g$^{-1}$ soil) in the microcosms after 3 days of incubation (Fig. 2a, ALONE), which corresponds to a 100-fold or more increase compared to their inoculated population sizes ($1 \times 10^5$ cells g$^{-1}$ soil). This demonstrated that the strains could grow at the expense of available resources within the soil microcosms. As expected, *E. coli* proliferated poorly and only increased its cell density by 10–12-fold within the soil (3–4 generations; Fig. 2a, ALONE), suggesting it found few nutrient niches to thrive. Over time all axenic populations slowly decreased in size suggesting some cell death occurred.

In contrast, when co-inoculated with GROWING NatComs or inoculated into STABLE NatComs all inoculant populations attained significantly lower population sizes than axenically in soil microcosms (Fig. 2a, Kruskal-Wallis test, *post hoc* Dunn pairwise tests,

Supplementary Data 1). The average inoculant population sizes here reached between $5 \times 10^4 - 2 \times 10^5$ cells g$^{-1}$ soil, depending on the NatCom state and inoculant (Fig. 2b), but remained relatively stable until the end of the experiment (Fig. 2a, approx. two months). The average growth and survival of inoculants was better in GROWING than in STABLE NatComs (Fig. 2b, c, d; see *P*-values in figure; Supplementary Data 1), but the population sizes of *E. coli* were the lowest among all inoculants and no different in GROWING or STABLE NatComs (Fig. 2b, c; Supplementary Data 1). Among the pseudomonads, *P. protegens* showed the highest mean population size and net population expansion (in comparison to the inoculated level; Fig. 2c, Supplementary Fig. 2). The mean population densities of all pseudomonads after two months in GROWING NatComs was higher than their initial inoculum, whereas those of *E. coli* and those in STABLE NatComs were not (although all strains were still detectable; Fig. 2d, Supplementary Fig. 2). These results support our hypothesis that the soil inoculants (all pseudomonads but not *E. coli*) were able to find more available niches for their establishment within a diverse soil resident community under GROWING conditions than in the background of an established STABLE community. The difference in inoculant population size in axenic (Fig. 2a, ALONE) vs. community-seeded microcosms (Fig. 2a, GROWING or STABLE) indicated that on average only 0.45 (STABLE) – 1.2% (GROWING, mean of the ratios at all time points) of the potential nutrient niche for the (pseudomonad) inoculants is realised within a taxonomically diverse resident soil community. Given that inoculant population sizes even under axenic conditions attained only ca. 10% of the measured community densities in the microcosms (Fig. 2a), this suggests that, in first instance, it is not the lack of spatial niches but nutrient niche competition that limits inoculant expansion. Furthermore, these results showed that pseudomonads have better colonization success in soil than a poorly soil-adapted strains such as *E. coli*. However, they did not attain cell densities higher than two times the inoculum size.

## Creation of a specific nutrient niche favors inoculant establishment in resident communities

To test whether inoculant outgrowth is indeed limited by nutrient niche competition and not lack of spatial niches or other, we exploited the capacity of one of the inoculants (*P. veronii*) to metabolize toluene, which we could add as a selective carbon substrate (assuming that the ability of NatCom strains to metabolize toluene would be limited). Toluene was provided in the gas phase of the microcosm from where it could reach the cells in the soil pore aqueous phase by diffusion. We repeated then the experiments with GROWING and STABLE NatComs, but now exposed to toluene, either inoculated with *P. veronii* or not.

Supplementation of toluene had no statistically significant effect on the sizes of the STABLE NatCom (Fig. 3a, $P_{STABLE} = 0.35372$) but significantly increased community sizes in GROWING NatComs by an average of 1.5-fold ($P_{GROWING} = 0.00869$). This was unexpected but

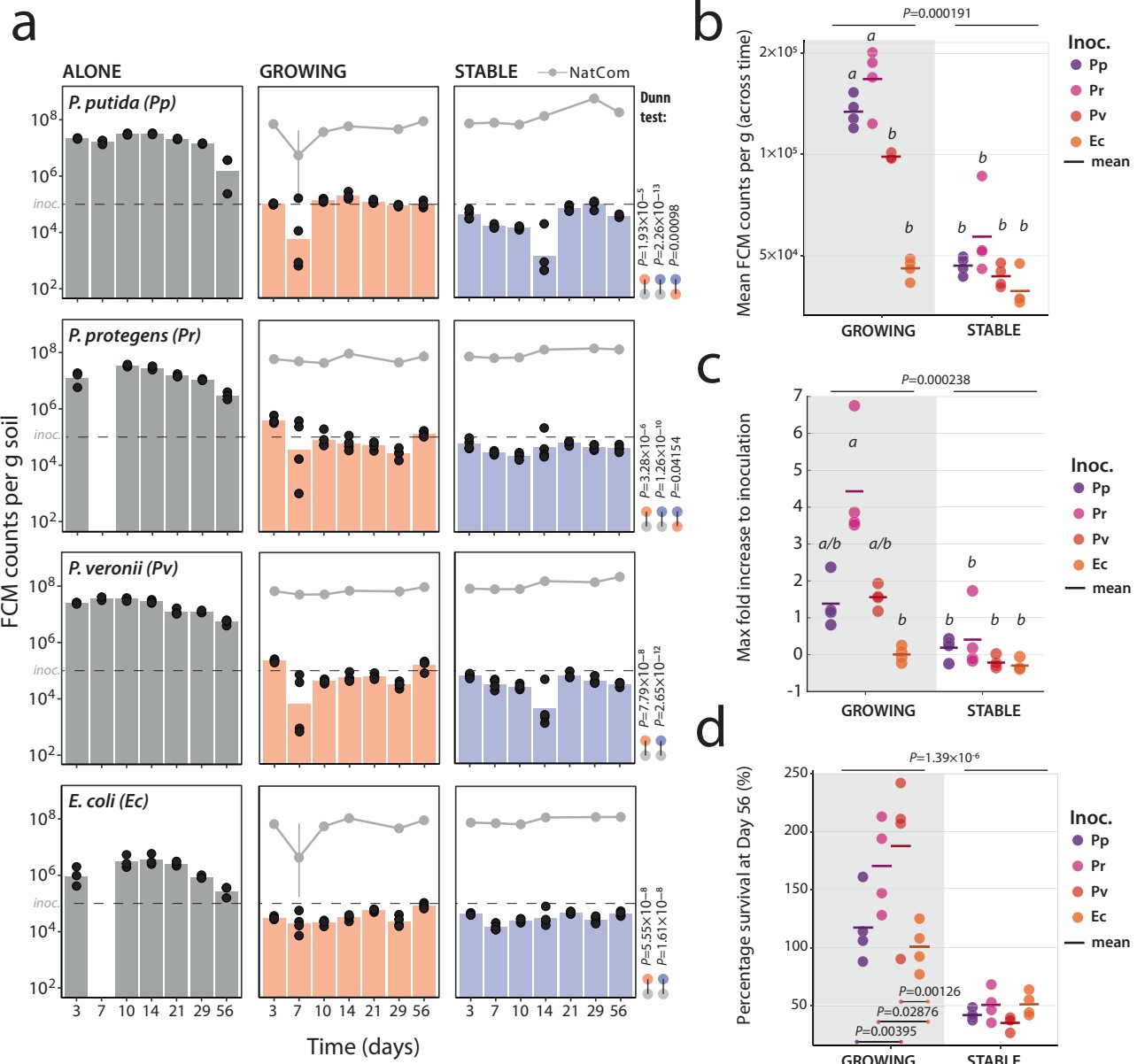

**Fig. 2 | Dependence of inoculant establishment on the growing state of resident communities. a** Bars show mean inoculant (*P. putida, P. veronii, P. protegens,* or *E. coli*) population sizes at each time point in sterile microcosms (ALONE, gray) and microcosms with GROWING (orange) or STABLE NatComs (blue). Black circles indicate individual biological replicate values (*n* = 4). Population sizes are expressed as flow cytometry (FCM) cell counts (mCherry fluorescence) per g of soil. Gray lines in GROWING or STABLE subplots connect the mean total community size measurements (SYBR Green I staining) in the same samples (vertical lines representing ± one standard deviation of the mean). *P*-values of Kruskal-Wallis test *post hoc* Dunn comparisons are indicated on the right of each group of subplots with colors indicating the comparison pair. Dashed lines indicate the calculated inoculum population size. **b** Average inoculant (Inoc.) population size in GROWING (grey background) or STABLE NatComs during the entire experiment (i.e., mean of all sampling time points, except T₀; *n* = 4 biologically independent microcosms). Individual replicate values are indicated by dots, with colors according to the key, and the replicate means shown by a line. *P*-values refer to differences among GROWING and STABLE NatComs by Kruskal-Wallis testing (letters or individual *P*-values grouping specific inoculant effects, from *post hoc* testing at <0.05). Pp, *P. putida*; Pr, *P. protegens*; Pv, *P. veronii*; Ec, *E. coli*. **c** As for (**b**), but for the maximum observed (any time point) fold-difference of inoculant density compared to T₀ (max flow cytometry counts – counts at start, divided by the counts at start). A value of zero indicates no difference to start. **d** as (**b**) but for the percent inoculant survival after two months (as the ratio of inoculant population size after two months divided by the initial inoculum size). *P*-values as in (**b**).

may be a consequence of some background growth by NatCom members on toluene, or of the dramatic disturbance in growth of most of the NatCom taxa (Supplementary Fig. 3), which changed the overall nutrient utilization patterns. In contrast, GROWING NatComs exposed to toluene but inoculated with *P. veronii* further increased the total community size by 1.8-fold, even at later time points, which was more than expected from the *P. veronii* population itself (Fig. 3a, b). Inoculation with *P. veronii* did not statistically

significantly change the STABLE toluene-exposed community size (Fig. 3a, *P* = 0.51487). As intended by the provision of the selective carbon substrate, *P. veronii* attained 100–200-fold larger population sizes in the presence of toluene compared to unamended microcosms, irrespective of being co-inoculated with GROWING or inoculated into STABLE NatComs (Fig. 3b, Kruskal–Wallis *P* = 2.2 × 10⁻¹⁶, *post hoc* Dunn test; Supplementary Data 1). Eventually, the *P. veronii* populations declined in the presence of toluene but still

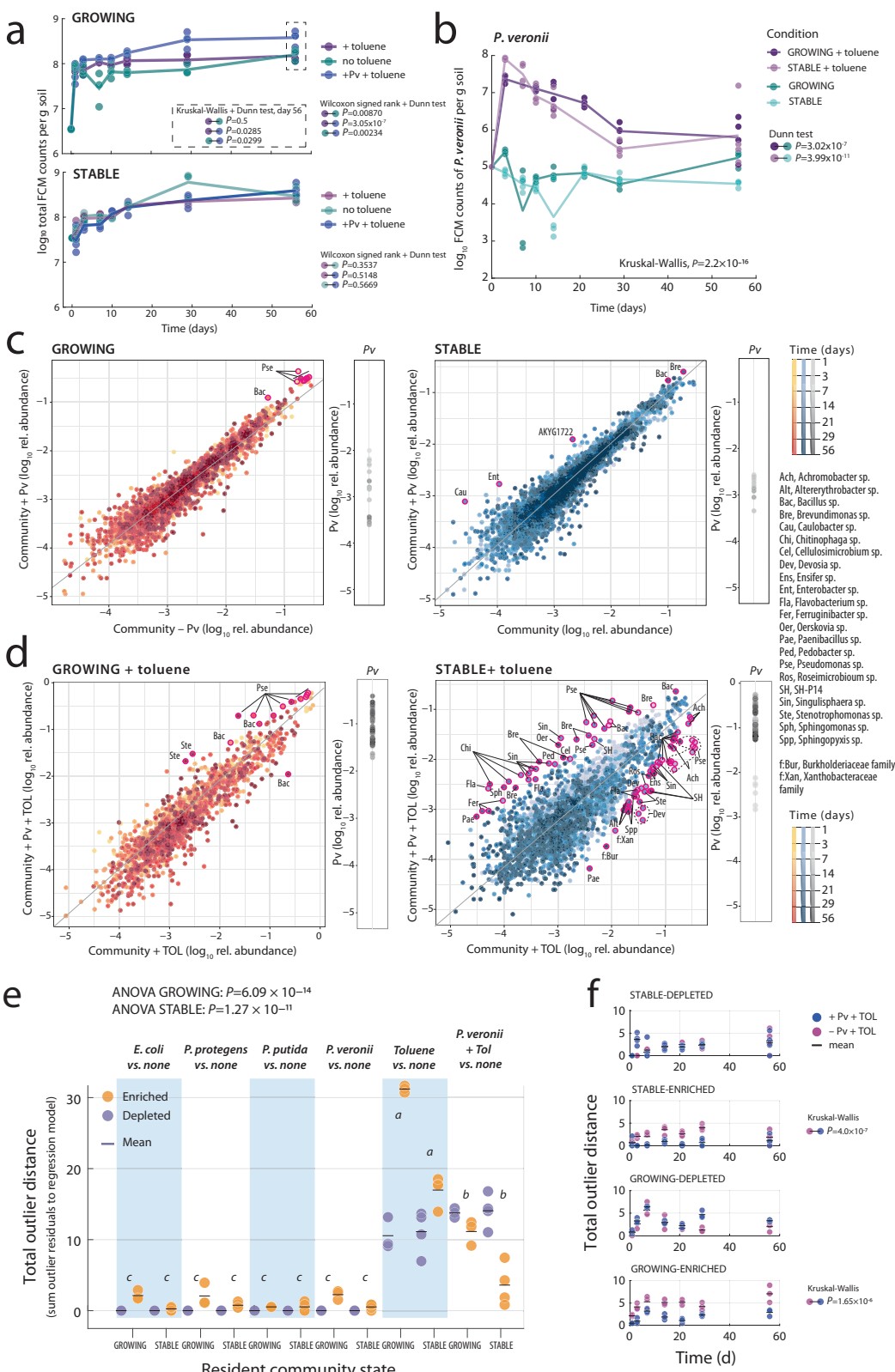

maintained higher levels than in its absence (Fig. 3b). This experiment thus demonstrated that the proliferation of an inoculant is significantly improved when it finds a selective nutrient niche. This also suggested that it is effectively the absence of a selective nutrient niche and competition for shared nutrients that limited the inoculant's development in the unamended NatCom microcosms, and less a lack of spatial niches.

Considering that in toluene amended microcosms *P. veronii* comprised up to 10%–20% of the total community size, we hypothesized that this may have caused secondary effects on resident populations. We thus compared paired taxa abundances in the absence or presence of *P. veronii*, per treatment, and over time (e.g., Fig. 3c, d). In the absence of toluene, either in GROWING or STABLE NatComs, inoculation of *P. veronii* caused very few taxon outliers to appear

**Fig. 3 | Effect of a toluene selective nutrient niche on growth and survival of *P. veronii* within resident soil communities. a** Community sizes in presence or absence of toluene, with or without *P. veronii* (Pv). Dots show replicate total flow cytometry (FCM) cell counts per g soil ($n = 4$ microcosms; 'no toluene' data reproduced from Fig. 1e for clarity) with lines connecting the means. *P*-values from Wilcoxon sign rank and *post hoc* Dunn tests (two-sided), in pairs as per the colors. Kruskal-Wallis *post hoc* Dunn test of the effect of inoculant on the GROWING NatCom sizes at day 56. **b** *P. veronii* population development within GROWING or STABLE NatCom exposed to toluene or not. (*P. veronii* without toluene reproduced from Fig. 2a for clarity). *P*-value from Kruskal-Wallis test for the difference between toluene- or non-treated samples. *Post hoc* Dunn *P*-values test differences of *P. veronii* population sizes at day 56 as a function of toluene treatment between GROWING or STABLE NatComs. **c, d** Deviations of amplicon sequence variant (ASV) relative abundances in *P. veronii* inoculated vs. non-inoculated microcosms (GROWING or STABLE), without (**c**) or with exposure to toluene (**d**). Dots represent time-paired $\log_{10}$-transformed relative abundances of the same ASV across all biological replicates (*Pv* abundances shown in subplot) and treatments. The diagonal trendline is the log-linear regression of all GROWING or STABLE comparisons, excluding toluene. Magenta circles emphasize outlier ASVs (genus abbreviation according to list on the right), whose relative abundance is more than five-fold different than the standard deviation of the residuals of the log-linear regressions. **e** Total outlier distances (dots are individual replicate values, $n = 4$ experiments), per treatment or inoculant in comparison to no treatment, as the sum of all outlier residuals to the regression models, separated for enriched or depleted taxa. ANOVA tests the complete GROWING or STABLE data sets, with letters indicating significantly different groups in *post hoc* multiple testing ($p < 0.05$). **f** As in (**e**), but for the comparison of NatComs exposed to toluene in presence or absence of *P. veronii*, and separated per time point (*P*-values from two-sided Kruskal-Wallis signed rank test).

(Fig. 3c; outliers defined as having a relative abundance of more than five times the standard deviation of the residual dispersion of linear regression of all $\log_{10}$-transformed community data, not including toluene; Supplementary Data 2). The absence of clear effects was not surprising given the relatively low attained population size of *P. veronii* in these microcosms (<1%; Fig. 3b).

In contrast, toluene exposure caused dramatic shifts in taxa abundances in both STABLE and GROWING NatCom, and both in presence or absence of inoculated *P. veronii* (Supplementary Fig. 3, Supplementary Data 3). This may be the consequence of replacement of toluene-sensitive taxa with more resistant ones. In comparison, however, GROWING NatComs exposed to toluene and inoculated with *P. veronii* were notably enriched for other Pseudomonas, a number of Stenotrophomonas and Bacillus strains (Fig. 3d, full list in Supplementary Data 3). Inoculation of *P. veronii* into STABLE NatComs exposed to toluene yielded a larger variety of enriched and depleted taxa, among which, notably, again several Pseudomonas strains (Fig. 3d, Supplementary Data 3). Quantified across all conditions and treatments as the respective sums of the outlier residual distances to the regression model, none of the inoculants alone caused particular secondary taxon enrichments or depletions (Fig. 3e, total outlier distances <5). In contrast, toluene exposure in presence or absence of *P. veronii* led to tenfold higher total outlier distances, with significant enrichments and depletions both in GROWING and STABLE NatComs (Fig. 3e, ANOVA, $P = 1.27 \times 10^{-11}$, $6.09 \times 10^{-14}$, Supplementary Data 1). Outlier distances increased in the first week of toluene exposure, particularly in GROWING communities, suggesting toluene directly affects growth of community members (Fig. 3f). Fewer taxon outliers appeared in presence of *P. veronii* in both GROWING and STABLE NatComs exposed to toluene (Fig. 3e, ANOVA *post hoc* test; Supplementary Data 1). The inoculation and growth of *P. veronii* coincided with a temporal increase in depleted outliers of STABLE communities, and at all time points diminished the magnitude of enriched outliers in both STABLE and GROWING communities (Fig. 3f, $P = 1.65 \times 10^{-6}$, $4.0 \times 10^{-7}$), suggesting it has a moderate restoration effect on toluene-induced changes. Interestingly, the *P. veronii* population size exposed to toluene declined less rapidly when resident NatCom was present (irrespective of GROWING or STABLE condition), compared to when it was growing axenically in microcosms (Supplementary Fig. 3). These results thus indicated that addition of toluene provides a selective carbon substrate that can overcome nutrient niche limitations on *P. veronii* growth in presence of NatComs. However, toluene also exerts toxic effects on the NatCom taxonomic composition and whereas *P. veronii* inoculation does lead to on average higher productivities (Fig. 3a), it only partly alleviates the global disturbance effects on the community composition (Fig. 3e, f; notice that in this experimental setup the toluene supply was kept constant over time).

## Inoculants lose productivity but favor growth of soil community members in random paired assays

Because the microcosm results pointed to a central role of nutrient niche availability in inoculant proliferation, we next aimed to better understand the nature of potential (competitive) interactions between introduced inoculants and resident community taxa. To study this experimentally, we employed a method where single inoculant cells are randomly encapsulated and incubated with individual isolated soil cells within 40–70 μm agarose beads (the Poisson encapsulation process yields beads with on average 1–2 cells of either inoculant, resident soil cell or both; Fig. 4a)[31]. In contrast to the work above with standardized NatComs, we here used bacterial cells freshly washed from their natural soil matrix, thereby expanding the range of taxon-inoculant combinations being explored. Paired bead inoculant-community mixtures were incubated with different substrate mixtures to mimic growth in soil, including a similar aqueous organic compound fraction extract of the same soil deployed for the cell isolation (i.e., *soil extract*), a mixture of 16 carbon substrates thought to promote general growth of soil bacteria[34], or toluene. We hypothesized that, because of the proximity of founder cell pairs inside the beads, growth interactions would lead to deviations in the average microcolony size distribution of inoculant or soil taxon compared with either member growing individually.

Paired growth was quantified by estimating the size of fluorescent microcolonies inside beads at different incubation times. Inoculant colonies were distinguished from the fluorescently stained soil taxa courtesy of their mCherry-fluorescence labels. As example, the average size of encapsulated *P. veronii* microcolonies, incubated with soil extract as the sole carbon and nutrient source, increased more over time if *P. veronii* was incubated alone (Fig. 4a, Pv ALONE) compared to beads where *P. veronii* was paired together with soil community (Fig. 4a, Pv WITH SOIL CELLS IN MIX, $P = 0.0005$, Fisher's two-tailed distribution test). Inversely, soil cells appeared to benefit from incubation with *P. veronii*, as their average microcolony size increased when *P. veronii* was present as a partner (Fig. 4a, SOIL CELLS ALONE vs. SOIL CELLS WITH Pv IN MIX). This was not a result of differences in medium conditions because incidental beads in the inoculant-partner incubations with only *P. veronii* (Fig. 4a, Pv ALONE IN MIX) showed similar average growth as the separate control incubations.

The same pattern was observed under most other tested nutrient conditions and with each of the four inoculants (Fig. 4b). This suggests that all the inoculants (i.e., *P. veronii*, *P. protegens*, *P. putida*, and *E. coli*) transform primary substrates into metabolites or otherwise increase local nutrient availability, leading to growth benefit of other soil cell taxa in proximity (i.e., within the same bead). The process is disadvantageous for the inoculant itself as it reduces its own productivity. To show this more clearly, we selected beads from different time points containing exactly one inoculant and one soil cell taxon microcolony (Fig. 4c). This enabled us to detect shifts in paired

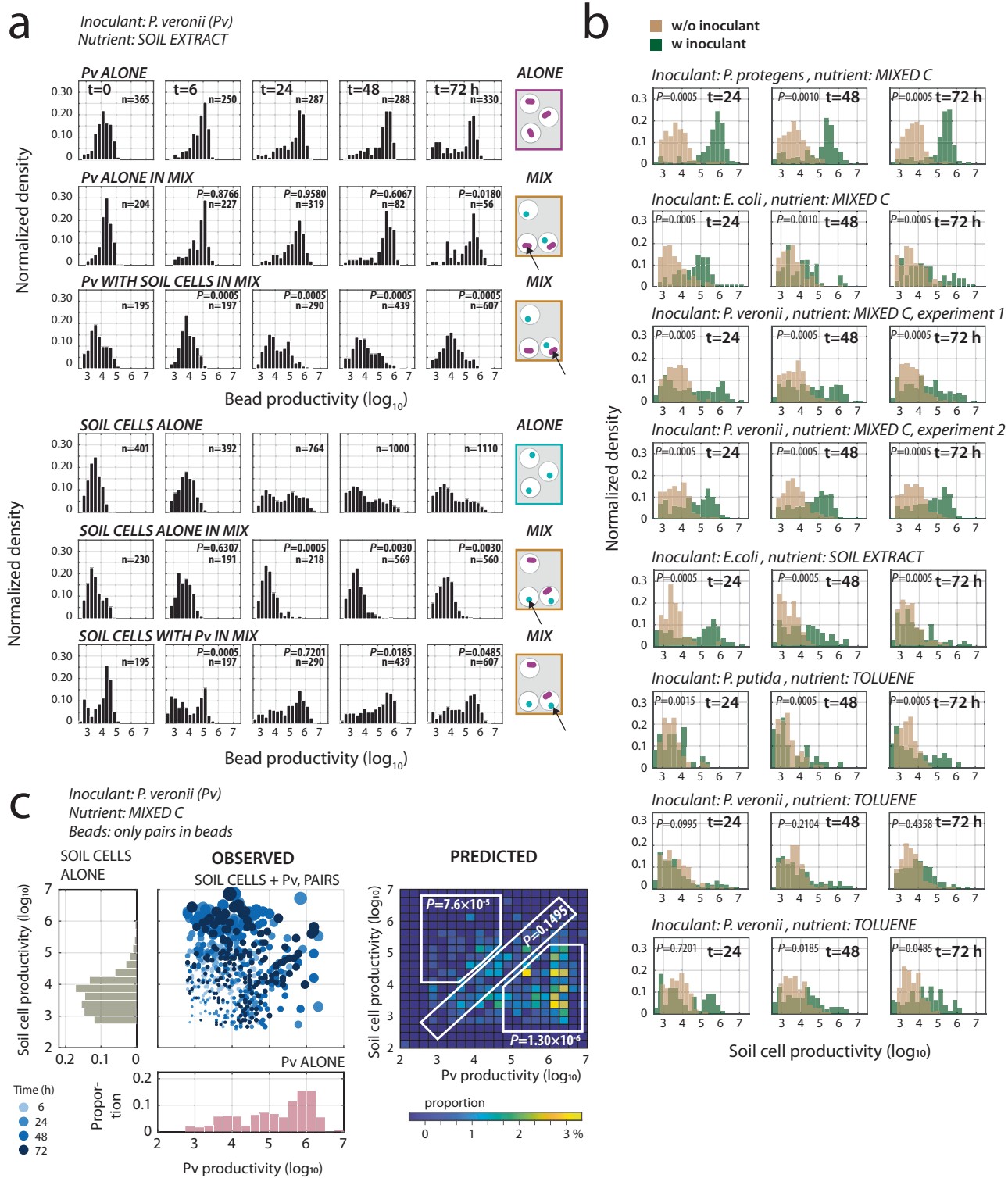

productivities compared to the productivities expected from growth distributions of each member alone if they were indifferent to each other (Fig. 4c, *PREDICTED*). The experimental results clearly show a stronger than expected growth of soil taxa and consequently reduced growth of *P. veronii* in paired growth tests ($P = 7.6 \times 10^{-5}$, two-tailed $t$-test, $n = 3$ experimental and 5 simulation replicates). We consistently observed similar outcomes across all four inoculant strains and in each growth condition, indicating the higher-than-expected growth of soil cells in paired beads with inoculants (Supplementary Fig. 4). Thus, these findings demonstrated that, on

average, all inoculants lose in substrate competition when in proximity of a soil cell, from which the latter can not only profit but more so than if growing alone. This also indicates that it is not only the direct loss of available nutrient niches, e.g. by faster-growing taxa, that limits inoculant growth but also the loss of competitiveness during nutrient niche transformation (i.e., *competitive loss by facilitation*). Both effects help to explain why the inoculants established much more poorly in soil microcosms co-inoculated or precolonized with NatComs compared with axenic microcosms (Fig. 2a, GROWING and STABLE vs. ALONE).

**Fig. 4 | Paired productivities of inoculants with random resident soil cells.**
**a** Growth of either bead-encapsulated *P. veronii* (Pv) or soil cells alone, or of paired mixtures (e.g., Pv with soil cells, 1–2 random founder cells at start) in soil extract medium. Note the illustrations on the right explaining how mixed beads can by chance have true pairs of inoculant and soil cells or contain single cells of either of them. Plots show normalized histograms of bead productivity: microcolony sizes imaged from epifluorescence microscopy, expressed as the $\log_{10}$-value of the Syto9-observed pixel area × its mean fluorescence intensity. n, number of analyzed beads. *P*-values from Fisher's exact test (two-sided) on distribution differences in comparison to ALONE. **b** As for (**a**) but for different inoculants and media conditions and only showing the comparison of soil cells alone (in beads in the mixture, light brown) and in beads with soil cells and inoculants (dark green). *P*-values from Fisher's exact test (two-sided). **c** Paired productivity plot of beads with only a single

*P. veronii* and a single soil cell microcolony (OBSERVED), versus the distribution of bead-growth of *P. veronii* (Pv, salmon) or soil cells alone. Pv and soil cells alone summed from time points 24, 48, and 72 h. Circles are proportional to the sum of the measured microcolony sizes (colors follow color key). The heatmap (PREDICTED) shows the expected paired bead summed productivities (in percentage, as per the color scale) from the individual measured microcolony sizes (i.e., Pv and soil cells ALONE) for the same number of beads as analyzed by microscopy. *P*-values correspond to the two-tailed *t*-test comparison of the variation of the total measured paired productivities inside the three regions ($n = 3$; 24, 48, and 72 h) with that in the simulations ($n = 5$). The upper left region shows higher soil cell productivity than expected, the lower right region shows lower inoculant productivity than expected, and the diagonal shows the same productivity for both microcolonies in a pair.

To better understand whether the competitive loss by facilitation is soil taxon specific or general, we analyzed changes in taxa abundances as a function of growth conditions and inoculant using 16 S rRNA gene amplicon sequencing of bead mixtures with or without inoculants (focusing only on *P. veronii* and *E. coli*; Fig. 5). DNA isolated from beads after 48 h contained between on average 71–205 unique ASVs ($n = 3$, threshold > 10 reads, Supplementary Data 4). Comparatively speaking, the bead-grown communities (subtracted from inoculant reads) showed clear substrate (soil extract, the solution of 16 mixed carbon sources or toluene) and inoculant type effects (Fig. 5a, Spearman correlation with ASVs collapsed to genus-level for comparison). Notably, soil extract as a sole nutrient source enabled growth of twofold higher taxonomic diversity than mixed C or toluene (Fig. 5b, ANOVA, mean *P*-value $1.66 \times 10^{-4}$; Supplementary Data 1, 4). Random pairing with *P. veronii* stimulated between 30–35 taxa on soil extract or mixed C, but in pairs with *E. coli* on mixed C only 6–15 taxa were significantly more abundant than in absence of inoculant (Fig. 5b and Supplementary Fig. 5, *mattest*, multiple test-corrected *P*-adj-value < 0.05 and $\log_2$-fold >2). In case of toluene as substrate in presence of *P. veronii* only 8 taxa were significantly enriched, among which other Pseudomonas genera but also Sphingobacterium, Brevundimonas, Stenotrophomonas and Comamonas strains (Fig. 5b, Supplementary Data 5). Most of the depleted taxa in the bead incubations were also poorly abundant, suggesting they did not manage to grow.

To compare effects of toluene exposure and *P. veronii* inoculation among bead-grown communities and NatCom microcosms, we collapsed the individual significantly differently abundant ASVs to Family level and scored the Family-level attributions as proportions of all ASVs in those Families (Fig. 5c). For example, one can see how toluene leads to enrichment in the Pseudomonadaceae both in bead and microcosm experiments (GROWING and STABLE; those being also relatively highly abundant), and to lesser extent in members of Xanthomonadaceae, Sphingobacteriaceae, or Caulobacteriaceae (Fig. 5c). At the level of the complete resident communities (inoculant counts removed), there is a strong separation of bead- from microcosm communities in nonmetric multidimensional scaling analysis (Fig. 5d; $P = 0.001$, $r^2 = 0.310$ by PERMANOVA), and a clear effect of toluene ($P = 0.001$, $r^2 = 0.191$). In contrast, the effect of *P. veronii* as inoculant is globally not significant (Fig. 5d, $P = 0.075$). Taken together, this comparison indicated that inoculants stimulated a variety of soil taxa when growing in proximity and given general nutrients (Fig. 4b), whereas toluene causes a more selective stimulation by *P. veronii* of similar soil taxa both in beads and microcosms (Fig. 5c). Such evidence underscores the notion that successful inoculant proliferation in soil is challenging when only general substrates are provided due to competitive loss or, depending on the perspective, carbon facilitation to others.

### Inoculant toluene metabolism triggers a variety of cross-feeding pathways in the resident community

Given the evident interactions taking place in the beads, we tried to delineate the potential cross-feeding network among resident soil

bacteria arising from inoculation. We focused here specifically on inoculation of *P. veronii* and exposure to toluene, as under these conditions the inoculant population could establish sufficiently such that potential effects on resident bacteria might be detected. We hypothesized that while toluene provided a specific growth advantage to *P. veronii* within the NatComs, its metabolism could indirectly facilitate the growth of other taxa, as suggested by both the encapsulation experiments (Figs. 4c and 5b, c) and microcosm studies (Fig. 3a, d). Here, we took advantage of a previously conducted study where *P. veronii* was inoculated into two types of natural soils, Silt and Clay, and into a contaminated control soil from a former gasification site (Jonction)[35]. The use of varied, non-sterile soils and materials beyond standardized microcosms is important here to demonstrate the more general nature of *P. veronii* inoculation successes and its impacts on the soil resident community.

Total RNA isolated from microcosms containing each soil type at early and late time points (Fig. 6a) was subjected to metatranscriptomic sequencing, assembly, and annotation to quantify gene expression levels of the native soil taxa. *P. veronii* established in Silt and Jonction soils exposed to toluene, while Clay soils demonstrated higher resistance to inoculant establishment despite toluene addition (Fig. 6a). Soils where *P. veronii* was actively growing (Fig. 6a, Silt and Jonction) also showed higher abundances of transcripts for ribosomal proteins, which indicates increased activity and growth of the resident community (Fig. 6b). Notably, uninoculated but toluene-exposed Silt resident communities had higher abundances of transcripts for ribosomal proteins but only later in the incubation. Resident microbiota transcripts associated with aromatic compound metabolism were enriched under toluene exposure in presence of inoculated *P. veronii*, particularly for pathways linked to known metabolites of the *P. veronii* toluene degradation pathway (Fig. 6c, d; I, II and III; Supplementary Fig. 6). Such enrichments were particularly prevalent in Silt microcosms compared to their corresponding uninoculated toluene-free or inoculated and toluene-exposed controls (Fig. 6d). Toluene exposure in the absence of *P. veronii* also provoked an increase in transcript abundance of aromatic compound degradation pathways but generally at the later sampling point, suggesting some growth of native toluene degrading strains. Jonction, as expected for an already contaminated soil, carried high transcript levels of a higher functional diversity of genes for aromatic compound metabolism in comparison to Silt or Clay (Supplementary Figs. 6 and 7). These transcripts could be assigned to close relatives of *Immundisolibacter cernigliae* and *Rugosibacter aromaticivorans* (Supplementary Fig. 8), two known degraders of aromatic compounds[36,37]. Transcripts related to aromatic compound metabolism were generally low in the Clay microcosms, probably because *P. veronii* did not proliferate well and no secondary effects had taken place (Fig. 6a). Interestingly, the sum of transcripts in the resident soil microbiota for the exploitable toluene degradation products followed a log-linear correlation with their overall growth state, as estimated from transcripts for ribosomal proteins (Fig. 6b). For a small number of increased aromatic compound metabolism

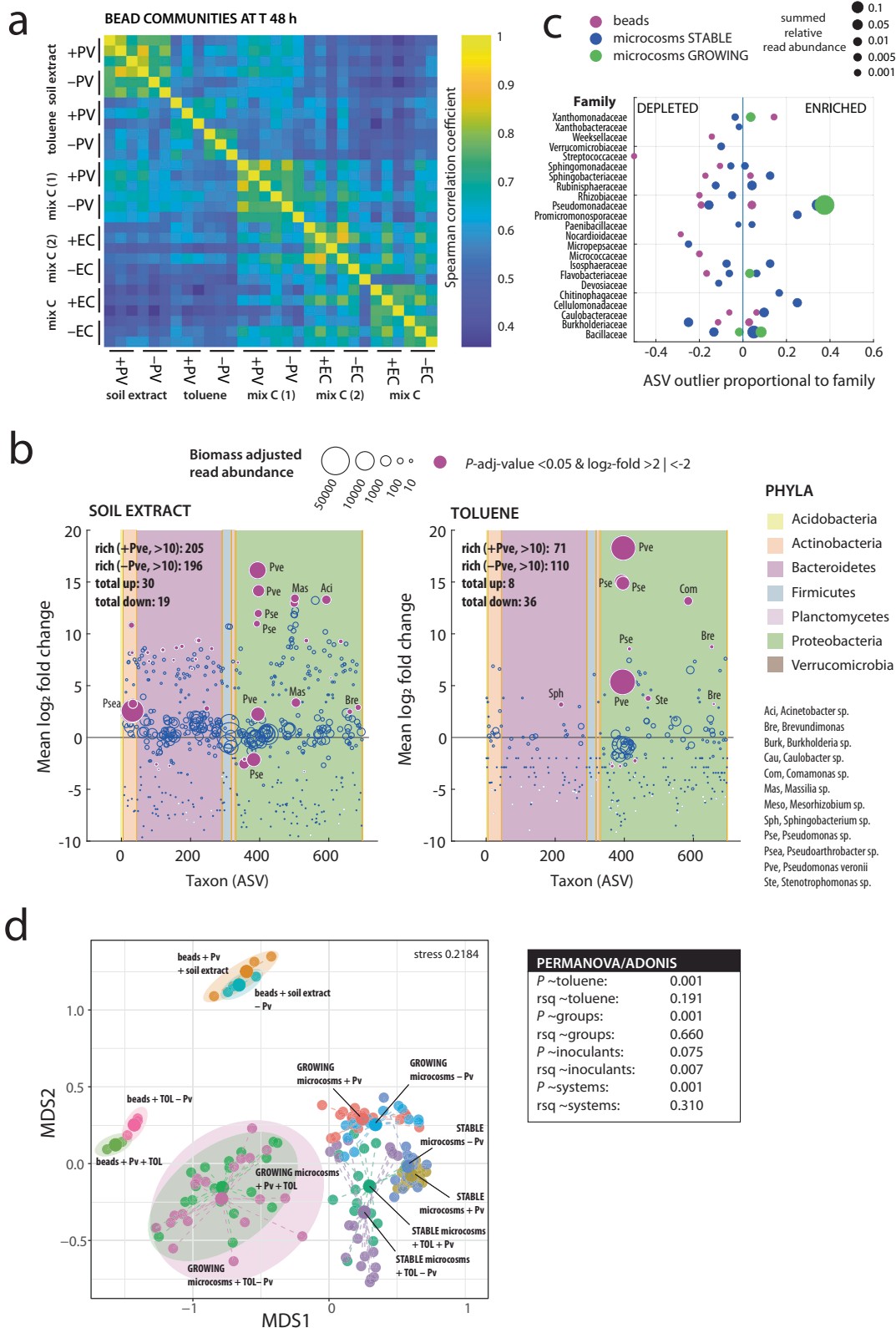

transcripts we traced the potential source organism (Supplementary Fig. 8). As expected, identified taxa were already enriched in Jonction but other stimulated taxa in Silt and Clay became enriched following toluene and *P. veronii* exposure in agreement with the observed Nat-Com and bead community stimulations (Fig. 5c). In summary, these results show that the metabolism of toluene by *P. veronii* can elicit a cascade of cross-feeding pathways among resident soil bacteria.

## Discussion

Microbiome interventions based on strain inoculations are frequently frustrated by the poor proliferation of the inoculant and, thus, an insufficient display of their intended function[7,12,13]. To better understand the underlying ecological conditions and mechanisms leading to poor inoculant proliferation, we systematically studied the potential of available nutrient niches on the establishment of a variety of inoculant

**Fig. 5 | Diversity changes in inoculant-soil cell bead-encapsulated communities as a function of growth condition and in comparison to soil microcosms.** **a** Spearman correlation of taxa relative abundances in inoculant-soil or soil alone bead-encapsulated communities after 48 h growth on different substrates, identified by amplicon sequence variant (ASV) but collapsed to Genus level (inoculant reads removed; PV, *P. veronii*; EC, *E. coli*). Mix C, mixed carbon substrates. Each set has three replicates. **b** Mean $\log_2$ -fold change of absolute read abundances of individual ASVs (corrected for imaged biomass differences) in bead-encapsulated communities with or without *P. veronii* and incubated with soil extract or toluene after 48 h. Background colors show phyla attribution as per the color key on the right. Dots indicate mean values from biological triplicates with a circle size proportional to the absolute read abundance in the data set. Rich, ASV richness (only ASVs with more than 10 reads retained). Magenta dots denote significantly different taxon abundance changes ($\log_2$-fold >2 or <−2; *mattest* adjusted *P*-value < 0.05, two-sided, multiple test correction) with abbreviations explaining the corresponding Genus. **c** Comparisons of outlier taxa (summed as proportion of ASVs within the attributed Families) between soil-paired encapsulations with or without *P. veronii* and toluene exposure, and toluene-exposed NatComs. Circle size proportional to the summed relative read abundance of the outliers. **d** Non-metric multidimensional scale analysis of bead-encapsulated paired *P. veronii* (Pv)-soil cell communities and NatComs (GROWING and STABLE), exposed or not to toluene (TOL) or soil extract. Group centroids are represented by larger circle diameters, and highlighted further by common colors or explanatory arrows. PERMANOVA/ADONIS *post* testing of treatment effects. Rsq, r-squared correlation coefficient.

strains in soil microcosms containing taxonomically diverse resident soil microbiota. Our studies benefitted from the reproducible culturing of taxonomically complex soil microbiomes, which enabled us to contrast the proliferation of inoculants concomitant to growing soil microbiota with their invasion into a stabilized precultured soil microbiota background. By comparing the growth of inoculants axenically in the same soil microcosm conditions we found that only around 0.45–1.2% of the potential nutrient niche available to the inoculant is free in the presence of NatCom (Fig. 2a). The available nutrient niche was four times greater when the inoculant was co-inoculated with NatCom than if it was introduced after colonization by NatCom (Fig. 2b, c). This was only the case for pseudomonad inoculants and not for the non-soil strain control *E. coli*, which, as expected, survived very poorly when inoculated into NatCom. Given that the starting density of inoculant ($1 \times 10^5$ cells $g^{-1}$) in the case of GROWING community was approximately one-tenth of the estimated resident Proteobacteria proportion (ca. 60% of $2 \times 10^6$ cells $g^{-1}$ soil, Fig. 1c, e) it is unlikely that these (opportunistic and fast-growing) Pseudomonas inoculants were outcompeted by faster consumption of primary growth substrates by NatCom taxa. Rather, as highlighted by random paired bead growth assays on various carbon substrate mixtures, inoculants lost productivity during growth (Fig. 4b). We call this effect *competitive loss by facilitation*, which has not been acknowledged as such before in the context of inoculation or microbiome engineering[38]. The findings from the inoculant-paired bead communities that different taxa become enriched depending on substrate (Fig. 5b, c), as well as our metatranscriptomic data of the specific case of inoculation of *P. veronii* and addition of toluene as a selective substrate (Fig. 6d), suggest that the underlying reason for the inoculant's competitive loss is leakage of metabolites and their profitable uptake by soil bacteria in the immediate vicinity (i.e., the facilitation). However, since we could not analyze behaviour of isolated inoculant-soil taxon pairs separately, we cannot exclude that secretion of growth-inhibitory substances by soil taxa can also be a factor in decreased inoculant growth.

Diversity has frequently been suggested as a key factor controlling the establishment of new species in resident microbiomes[5,39–41], whereas other studies have instead emphasized the importance of community productivity[19,42]. Both factors are inherently related, given that productivity reflects and depends on community composition within the resource richness of the habitat[43]. The more important underlying determinant of the community composition effect in this context, however, seems to be niche availability[18–20]. Growing, habitat-adapted, taxonomically diverse communities can be expected to deplete nutrient and occupy spatial niches, which limits the proliferation of incoming species (i.e., pathogens or inoculants)[2,21,24]. Habitats in quasi-steady state such as soil will, therefore, only have very restricted remaining nutrient availability when their resident community has reached stationary phase. We tested this effect on nutrient niche availability directly by maintaining the same taxonomically diverse NatCom in two system states: one with low (GROWING) and the other with high (STABLE) initial biomass. Indeed, the GROWING NatCom expanded six times more than the STABLE community, reflective

of the lower nutrient availability after colonization by the resident community. Similarly, also (three of the four) tested inoculants (the three pseudomonads) established better in the GROWING than the STABLE NatCom state (Fig. 2b–d). Their establishment was modest in comparison to the inoculant's axenic growth in the same habitats, suggesting that it is not so much the absence or occupation of spatial niches but nutrient niches that limit inoculant growth. The poor proliferation in soil of *E. coli* is likely due to its general inability to exploit soil ecological niches. By providing a selective nutrient niche (i.e., toluene) within the same background we achieved two orders of magnitude higher inoculant growth, demonstrating that niche (un)availability and competition control inoculant proliferation. The concept of a unique niche for inoculant growth has been understood for infant gut succession and can be exploited by synbiotic supplements[44,45], whereas recent work (which employed a similar experimental system) also demonstrated how nutrient provision in the plant rhizosphere can build a specific inoculant niche[46]. Our results now show it is applicable to soil microbiota interventions.

Although nutrient niche availability explained part of the inoculant's fates in the soil communities, we also investigated potential biological interactions, such as substrate competition or metabolite cross-feeding, which have been considered by others as crucial for invader establishment[5,7]. Surprisingly, we found that growth of all four inoculants was decreased in paired co-cultures with randomized soil bacteria inside micro-agarose beads (Fig. 4), whereas growth of the soil partner on average was increased. Rather than substrate competition this effect is indicative of facilitation, by which the inoculants lose productivity in facilitating the growth of the partner[47]. Although the process of bead pairing forces partnerships that, possibly, occur very infrequently in the soil habitat, it was interesting to see that inoculants are not necessarily in competition with phylogenetic close kin (e.g., *P. veronii* with other pseudomonads), except in case of toluene as substrate. Thus, our findings suggest that by facilitating more generally the growth of other soil bacteria taxa, inoculants diminish their own population expansion. Specifically for the case of thriving *P. veronii* in habitats exposed to toluene, both measurable increases of selective gene expression for aromatic compound metabolism in resident bacteria and enrichment of taxa within the same family were found (Figs. 5c and 6). More generally, the metabolism of most bacteria results in leaking metabolites[48] that can become more broadly accessible to other cells in their vicinity, thereby benefiting their maintenance or growth and contributing to community diversity[49].

Our results demonstrated the importance of niche availability for inoculant proliferation and highlighted the consequences of facilitative metabolism on competitive outcomes. From the perspective of microbiome engineering or interventions it is important to learn the degrees of available control of a system such that intended taxonomic and/or functional changes can be achieved. This control may range from exploiting inherent and temporal available niches for growth to establishing selective (temporal) niches for one or more inoculants to thrive and exert their functionalities. Engineering soil microbiomes is particularly complicated by their inherent biotic and abiotic

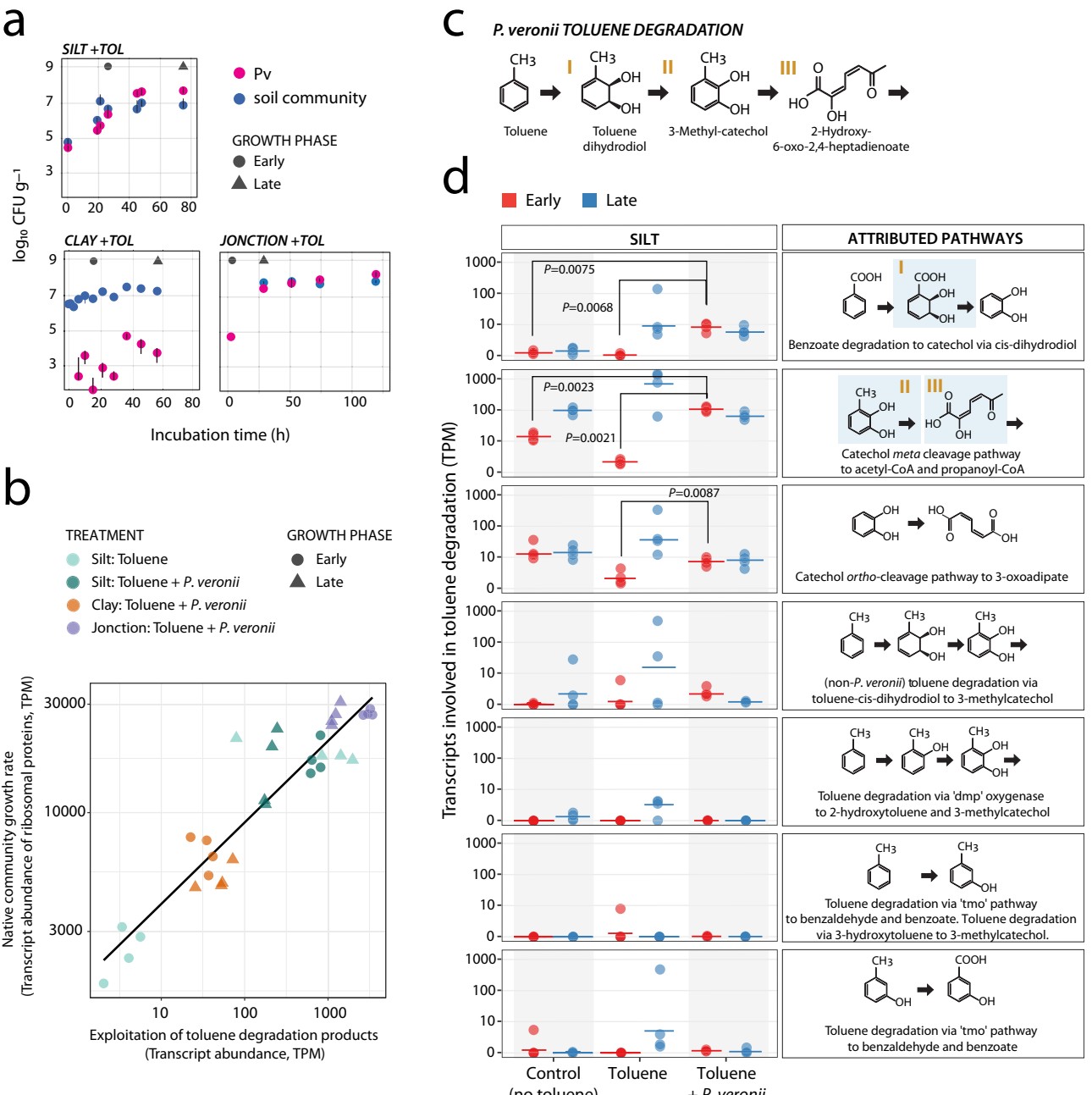

**Fig. 6 | Exploitation of toluene degradation products from *P. veronii* by soil microbiota. a** Survival or proliferation of inoculated *P. veronii* in Silt, Clay, or *Jonction* soils exposed to gaseous toluene (data replotted from ref. 35). Data points are the mean ± one standard deviation from n = 3 independent microcosm measurements of colony forming units of *P. veronii* (magenta) or resident soil microbiota (blue). Gray circle and triangle indicate time points for sampling of total community RNA (early and late time point, respectively). **b** Transcript abundances (TPM, transcripts per kilobase million, without *P. veronii* transcripts) of ribosomal proteins in the communities versus those of functions attributed to utilization of toluene or its metabolites (see **c**, **d**). **c** *P. veronii* toluene degradation pathway and major metabolic intermediates. Roman numerals correspond to highlighted intermediates in (**d**). **d** TPM-values of transcripts annotated to the aromatic compound metabolic steps on the right for Silt in three conditions tested at the early and late stages (as in **a**). Data points show individual values from n = 4 independent microcosm experiments with the line indicating the median. *P*-values correspond to *t*-test statistics comparison of indicated sample replicate measurements (two-sided; only shown if <0.05). For details of KEGG pathway attributions, see Supplementary Data 6.

complexity and spatial heterogeneity, which cannot be easily tuned by process parameters like, for example, in the engineered infrastructure of a wastewater treatment plant. Although several other factors can also influence population growth of inoculants, such as predation by protozoa[50] or phage killing[51], or biotic factors like pH[8], toxicity[52] or drought[53], our results indicate that nutritional niche engineering is a potentially exploitable mechanism to favor inoculant establishment. Engineered niches need not necessarily to consist of selective carbon

compounds but, potentially, could also be generated in the form of spatial niches or other limiting nutrients. *Inter alia*, our results also reflect a realistic bioremediation scenario where an inoculated bioremediation agent thrives thanks to the selective niche provided by a contaminating compound. Degradation of the contaminant can then simultaneously favor growth of other soil members, leading to the subsequent decline of the inoculant but restoration of the microbiome.

## Methods

### Soil inoculant strains

Four strains were selected as inoculants for growth and interaction studies with soil communities: *P. veronii* 1YdBTEX2, a toluene, benzene, *m*- and *p*-xylene degrading bacterium isolated from contaminated soil[27]; *P. putida* F1, a benzene-, ethylbenzene- and toluene-degrading bacterium from a polluted creek[28]; *P. protegens* CHA0, a bacterium with plant-growth promoting character as a result of secondary metabolite production[54]; and (motile) *E. coli* MG1655 (obtained from the *E. coli* Genetic stock center Yale; CGSC#8237)[55], as a typical non-soil dwelling bacterium. Variants of the four strains that constitutively express mCherry fluorescent protein were used, to facilitate their detection within resident soil communities by flow cytometry. *P. veronii* 1YdB-TEX2 and *P. protegens* CHA0 were tagged with mCherry (expressed under control of the $P_{tac}$ promoter) using a mini-Tn7 delivery system[56]. *P. putida* F1 was tagged with the same $P_{tac}$-*mCherry* cassette, cloned into and delivered by mini-Tn5 suicide vector pBAM[57]. *E. coli* MG1655 was tagged with the same $P_{tac}$-*mCherry* cassette but on plasmid pME6012[58].

### Culturing of NatCom soil microbial communities in soil microcosms

Inoculant proliferation was tested in sterile soil microcosms, either alone, or with or without precolonization by resident soil microbiota. The microcosms were prepared according to the procedure described in Čaušević et al. [30], by complementing dried, double-sieved, and twice autoclaved silt (obtained particle size 0.5–3 mm), with soil extract to a final gravimetric water content of 10%. Soil extract consisted of an aqueous phase extract from the same soil as used for preparation of the soil community. For NatComs this consisted of the 1–5 cm top layer of the Dorigny forest soil at the University of Lausanne campus[30]. Equal volumes of soil and tap water were mixed and autoclaved for 1 h and left to cool overnight. After decanting, the liquid fraction was further centrifuged and filtered (<0.22 μm) to remove soil and plant debris. The resulting solution was autoclaved a second time to ensure complete sterility. General parameters (e.g., pH, macronutrients) and the nature of the organic carbon fraction in the microcosms and the soil extract have been described before[30]. The soil microbiota was sourced from previously grown top soil microbial communities (NatCom) in the same type of reconstituted sterile soil matrix[30], which had been stored at room temperature (23 °C) for 1.5 years in Schott bottles (100 g material in a 500 ml bottle).

Soil microbiota were revived by transferring 11 g of the stored NatCom soil into 80 g sterile soil material reconstituted with 9 ml forest soil extract in a 500 ml capped Schott glass bottle (Fig. 1a, 50 replicates). Five microcosms were selected randomly for the Phase 1 analysis and the rest of the microcosm material was kept for Phase 2. We maintained 50 replicate bottles to avoid any growth differences resulting from upscaling. The microcosms were periodically mixed on a bottle roller and incubated at room temperature (23 °C) in the dark for 28 days to allow the growth, dispersal, and colonization of the NatCom throughout the soil material. The five selected microcosms were sampled at regular intervals to assess taxonomic composition by 16 S rRNA gene amplicon sequencing and cell density by flow cytometry (see below). After 28 days, the content of all inoculated microcosms was mixed and divided into two new sets of 28 microcosms. In one set (the STABLE state) the pooled and colonized soil material (100 g) was directly transferred to new, sterilized, and empty Schott bottles without any new addition of nutrients. In the second set (the GROWING state) the colonized material (11 g) was mixed with 80 g freshly sterilized soil matrix and 9 ml forest soil extract in a new bottle. This soil-to-soil dilution allowed a new phase of active community growth.

### Introduction of inoculants in soil microcosms

For inoculation into soil microcosms all pseudomonads were grown individually from frozen glycerol stocks in Lysogeny-Broth (LB, BD

Difco) supplemented with 25 μg mL$^{-1}$ of gentamicin (30 °C) and *E. coli* was grown at 37 °C in LB with 25 μg mL$^{-1}$ of tetracycline, to maintain the fluorescent marker. After 16 h culturing, cells were harvested by centrifugation, washed, and subsequently diluted in type 21 C minimal medium (MM; ref. 59) with 0.1 mM succinate to obtain a concentration of $10^7$ cells ml$^{-1}$. Four sets of four replicates each of STABLE or GROWING microcosms (see above) were then inoculated with either of the four strains to achieve a starting inoculant cell density of $10^5$ cells g$^{-1}$ of soil, while one set of four remained unamended to verify sterility. Inoculants were inoculated individually (ALONE) into sterile soil microcosms (4 replicates each, same microcosm material). A final two sets of four microcosms (with or without NatCom in either the STABLE or GROWING state) were amended with *P. veronii* and toluene or with toluene alone (see below). Following inoculation the microcosms were mixed on a bottle roller and incubated at 23 °C.

### Addition of toluene to microcosms

Toluene (Fluka Analytical) was introduced to the microcosms in the gas phase via 0.5 ml pure toluene held in a heat-sealed 1-ml micropipette tip, which was placed inside a sealed 5-ml tip for additional stability, carefully placed inside the microcosms. At each mixing and sampling step, toluene tips were removed from microcosms using sterile tweezers, the level of toluene was checked and replenished to 0.5 ml, if necessary, after which the tips were replaced once the content of microcosm was mixed.

### Extraction of cells from soil microcosms

Soil community size and composition was quantified using cells washed from the soil matrix at each time point. Microcosm material (10 g) was sampled using Sterileware sampling spatulas (SP Bel-Art) and transferred to a 50 ml capped Greiner tube, after which 10 ml sterile tetrasodium-pyrophosphate decahydrate solution (2 g l$^{-1}$, pH 7.5, Sigma-Aldrich) was added and the mixture vortexed for 2 min at maximum speed on a Vortex-Genie 2 (Scientific Industries, Inc.). Debris was allowed to settle for 2 min and the supernatant (cell suspension) was transferred to a new tube. This suspension was then used for cell enumeration by flow cytometry or colony forming unit counting, DNA isolation, and amplicon sequencing.

### Flow cytometry cell enumeration

A portion of the cell suspension (see above) was passed through a 40 μm nylon strainer (Falcon) to remove particulate material. Two 100-μl aliquots of filtrate were then mixed with equal volumes of 4 M sodium azide solution (Sigma-Aldrich) to fix the cells. Fixed samples were kept at 4 °C until processing with flow cytometry (within 1 week). Before flow cytometry measurement, one fixed sample was stained with SYBR Green I for 15 min in the dark (Invitrogen, following manufacturer's instructions) whereas the other remained unstained, allowing the estimation of background fluorescent particle content. Stained and non-stained suspensions (10 μl) were aspired on a CytoFLEX Flow Cytometer (Beckman Coulter) at the slow flow rate (10 μl min$^{-1}$). Phase 1 non-inoculated microcosms were used as controls for background noise coming from soil, which was subtracted from total counts of treated microcosms. The inoculants were detected and gated based on their mCherry tag (ECD-H signal) signal in the non-SYBR Green I–stained sample series (Supplementary Fig. 9).

### Colony forming unit counting

A 100-μl aliquot of soil cell suspension was serially diluted using tetrasodium-pyrophosphate decahydrate solution and 10 μl droplets (four technical replicates) of each dilution (from $10^0$ to $10^{-7}$) were deposited on R2A plates (DSMZ GmbH). The plates were left to dry for 10 min and then incubated at 23 °C in the dark. Colonies were counted after 3 days of growth using a stereo microscope (Nikon SMZ25), and the corresponding community number of colony forming units (CFU)

g$^{-1}$ soil was calculated from the cell extraction procedure and its dilutions.

### 16S rRNA gene amplicon sequencing

The remaining cell suspension (9 ml) was centrifuged in a swing-out rotor (Eppendorf A-4-62 Swing Bucket Rotor) at 4000 × g for 7 min to harvest the cells. The supernatant was discarded and cell pellets were stored at −80 °C. DNA was subsequently extracted from thawed cell pellets using a DNeasy PowerSoil Pro DNA Isolation Kit (Qiagen, as per instructions by the supplier). Final yields were quantified using a Qubit dsDNA BR Assay Kit (Invitrogen), and the purified DNA solution was stored at −20 °C until library construction. Each sample (10 ng DNA input) was then used to amplify the V3–V4 variable region of the 16 S rRNA gene, following the protocol by Illumina (16S Metagenomic Sequencing Library Protocol, https://support.illumina.com/documents/documentation/chemistry_documentation/16s/16s-metagenomic-library-prep-guide-15044223b.pdf).

Samples were indexed by using the Nextera XT Index kit (v2, sets A and B, Illumina) after which the DNA was again purified, pooled, and sequenced using a MiSeq v3 paired-end protocol (Lausanne Genomic Technologies Facility). Raw reads were analyzed using the Qiime2 platform on UNIX (version 2021.8)[60], and amplicon sequence variants (ASVs) were attributed to known taxa at 99% identity (operational taxonomic units, OTU) by comparison to the SILVA database (version 132).

### Paired inoculant-soil taxon growth assays in encapsulating agarose beads

Potential growth effects between inoculants and soil taxa were tested using random pairs of single cells encapsulated within 40–70 μm diameter polydisperse agarose beads[31]. Inoculants were precultured as follows. *P. veronii* and *P. putida* were grown on MM agar with toluene as sole carbon source provided through the vapour phase in a desiccator, as described previously[61]. A single colony grown after 48 h incubation at 30 °C was subsequently inoculated into 10 ml MM with 5 mM succinate as the sole carbon source and cultured for 24 h. *P. protegens* and *E. coli* colonies were cultured as described above on selective nutrient agar plates supplemented with 25 μg ml$^{-1}$ of gentamicin or 25 μg ml$^{-1}$ of tetracycline, respectively, and then transferred to liquid MM with 5 mM succinate. After 24 h growth, the cells were harvested from their precultures by centrifugation and resuspended in 10 ml MM. Cell suspensions were counted by flow cytometry and diluted to $2 \times 10^7$–$10^8$ cells ml$^{-1}$ for the bead encapsulation process. Soil microorganisms were washed and purified for each encapsulation experiment from four 200 g samples of fresh soil (characteristics and location as described previously[31]) using a similar procedure as described above for the NatComs. Purified cells were counted by flow cytometry and diluted in MM to $1 \times 10^8$ cells ml$^{-1}$ before encapsulation. Each of the inoculant or washed soil cell suspensions alone, or inoculant mixed in 1:1 volumetric ratio with the soil cell suspension, were then mixed with liquid low-melting agarose solution (37 °C) to produce 40–70 μm diameter agarose beads with a Poisson-average of two founder cells at start, using the procedure described previously[31]. Per condition and type of inoculant, two batches of cell-encapsulated beads were prepared in parallel, which were pooled and then split in three replicate tubes each containing 1 ml bead solution. The encapsulation procedure produced $1.2 \times 10^6$ beads per ml, with an estimated effective bead volume of 10% of the total volume of the liquid phase in the incubations.

### Culture conditions for bead-encapsulated cell pairs

Three different carbon source regimes were imposed on bead-encapsulated cells: toluene, mixed carbon substrates, or soil extract. Toluene was used as an example of an inoculant-selective substrate (for *P. veronii* and *P. putida*) and was provided by partitioning from an oil phase. We diluted pure toluene 1000× in 2,2′,4,4′,6,8,8′-heptamethylnonane (Sigma Aldrich) and added 0.2 ml of this solution to each vial with 1 ml bead suspension. A further 4 ml of MM was added to the vials for the incubation. Mixed carbon substrates (Mixed-C), and soil extract were used as diverse substrates for all inoculants and soil microbes. Mixed-C solution was prepared by dissolving 16 individual compounds (Supplementary Table 1) in milliQ-water (Siemens Labostar) in equimolar concentration such that the total carbon concentration of the solution reached 10 mM-C. These compounds have been used previously as soil representative substrates[34]. In the bead incubations, the Mixed-C was diluted to 0.1 mM-C final concentration in MM (5 ml total volume per vial) to avoid excessive growth of microcolonies inside the beads, which could lead to cell escape and their proliferation outside the beads.

Soil extract for agarose beads was prepared as follows. A quantity of 100 g soil (same origin as used for the soil community cell suspension[31]) was mixed with 200 ml 70 °C milliQ-water in a 250 ml Erlenmeyer flask and swirled on a rotatory platform for 15 min after which it was subjected to 10 min sonication in an ultrasonic bath (Telesonic AG, Switzerland). Soil particles were sedimented and the supernatant was decanted and passed through a 0.22-μm vacuum filter unit (Corning Inc.). The resulting soil extract (4 ml) was added directly to the 1 ml bead suspension in the vials. Triplicate vials per treatment and per inoculant-mixture were incubated at 25 °C with rotary shaking at 110 rpm to prevent sticking of the beads but avoid shearing damage.

### Sampling and analysis of cell growth in agarose beads

Encapsulated cell mixtures were sampled at regular time intervals (0, 6, 24, 48 and 72 h). For this, 10 μl of bead suspension was removed from the vials. Cells and microcolonies in the beads were stained with SYTO-9 solution and imaged with epifluorescence microscopy, as described previously[31]. Microcolony growth was quantified using a custom MATLAB (v. 2021b) image processing routine that segmented beads and microcolonies inside beads[62]. Inoculant cell colonies were differentiated from soil cells based on having both mCherry and SYTO-9 fluorescence, whereas soil cells displayed only SYTO-9 fluorescence. Growth was calculated as the product of SYTO-9 fluorescence area and mean fluorescence intensity for each detected microcolony[31]. Beads containing exactly one inoculant and one soil cell microcolony were selected to plot paired productivities. Productivities were compared to simulations (n = 5, implemented in MATLAB) of the expected paired productivity without any assumed interaction. First, a normalized probability distribution curve was build from the replicate observed growth in encapsulated beads of either the inoculant or the soil cells alone based on histograms as shown in Fig. 4c. The probability distribution curves were then five times independently randomly subsampled for the number of bead observations to build simulated pairs, with paired productivities summed per two-dimensional grid bin as displayed in the heatmap of Fig. 4c. Differences among observed and expected paired productivities were evaluated from the sums across three regions as indicated in Fig. 4c. Separate bead encapsulation series were conducted with or without *P. veronii* or *E. coli*, and with soil extract, mixed C or toluene, which were sampled after 48 h and frozen at −80 °C for DNA isolation and amplicon sequencing, as described above and elsewhere[31].

### Metatranscriptomic analysis

To better understand the impact of adding an inoculant and/or toluene on the native soil community we took advantage of previously conducted inoculation experiments of *P. veronii* in a variety of soil types from which total RNA had been purified[35]. These consisted of two uncontaminated soils (Clay and Silt), and one contaminated soil from a former gasification site named Jonction, as detailed previously in ref. 35. Soils had been exposed or not to toluene and inoculated with *P. veronii*. Soil microcosms had been sampled in an early or a late state

(see Fig. 6a; the exact timing roughly depending on observed growth of the inoculant population)[35]. Total purified RNA from the samples was depleted for bacterial ribosomal RNAs, reverse-transcribed, indexed, and sequenced on Illumina HiSeq 2500 or NovaSeq at the Lausanne Genomic Technologies Facility following a previously described procedure[35].

Sequencing reads from all samples were quality controlled by BBMap (v.38.71), which removed adapters from the reads, removed reads that mapped to PhiX (a standard added to sequencing libraries) and discarded low-quality reads (trimq=14, maq=20, maxns=1, and minlength=45). Quality-controlled reads were merged using bbmerge.sh with a minimum overlap of 16 bases, resulting in merged, unmerged paired, and single reads. The reads from metatranscriptomic samples were assembled into transcripts using the SPAdes assembler[63] (v3.15.2) in transcriptome mode. Gene sequences were predicted using Prodigal[64] (v2.6.3) with the parameters -c -q -m -p meta. Gene sequences from the GenBank entry of *P. veronii* (GCA_900092355) were downloaded and clustered at 95% identity, keeping the longest sequence as representative using CD-HIT[65] (v4.8.1) with the parameters -c 0.95 -M 0 -G 0 -aS 0.9 -g 1 -r 1 -d 0. Gene sequences predicted from assembled transcripts were used to augment the *P. veronii* database using CD-HIT (cd-hit-est-2d -c 0.95 -M 0 -G 0 -aS 0.9 -g 1 -r 1 -d 0). Representative gene sequences were aligned against the KEGG database (release April 2022) using DIAMOND[66] (v2.0.15) and filtered to have a minimum query and subject coverage of 70%, requiring a bitScore of at least 50% of the maximum expected bitScore (referenced against itself).

The 145 metatranscriptome samples were then mapped to the 246,873 cluster representatives with BWA[67] (v0.7.17-r1188; -a), and the resulting BAM files were filtered to retain only alignments with a percentage identity of ≥95% and ≥45 bases aligned. Transcript abundance was calculated by first counting inserts from best unique alignments and then, for ambiguously mapped inserts, adding fractional counts to the respective target genes in proportion to their unique insert abundances.

## Data processing and statistical analysis

Data processing, analysis of community composition, and statistical analysis were done using MATLAB (v. 2021b), GraphPad Prism (version 9.0.1) and R 4.0 (R Core Team, 2019) on RStudio (version 2022.2.3.492) using the following packages: *phyloseq*[68], *microbiome*[69], *MicrobiotaProcess*, *ggplot2*[70], *ggpubr*, *ggbreak*[71], *vegan*[72], *biomformat*[73], *tidyverse*[74], *reshape2*[75], *dplyr*, *Biostrings*[76], *scales*, *PMCMRplus*[77], *car*[78], *emmeans*[79], *rstatix*[80], *pairwiseAdonis*, and *RVAideMemoire*[81]. Chao1 values of Phase 1 samples (Day 0 to Day 23, different replicates) were compared with a Dunn test. Shannon values of Phase 1 samples were compared with a Kruskal-Wallis test and by pairwise comparisons using Dunn testing with Holm's *P*-value adjustment. Beta-diversity of Phase 1 community compositions (at species level) was analyzed using Unifrac distances with PCoA ordination (using *phyloseq* in R). Differences were analyzed using PERMANOVA (999 permutations) using the *adonis2* function, while data homogeneity was checked using *betadisper* function of the *vegan* package. Finally, pairwise differences between timepoints were investigated using *pairwise.perm.manova* from *RVAideMemoire* and *P*-values were adjusted using Holm's method. Phase 2 Chao1 and Shannon values of GROWING and STABLE were compared using two-way repeated measures ANOVA to investigate the effect of community state and time. ANOVA assumptions were checked with Shapiro-Wilk's normality test and Levene's test for homogeneity of variance. Outliers were investigated using the *identify_outliers* function of *rstatix* package. Data sphericity is automatically checked and corrected if necessary (via Greenhouse-Geisser correction) with the *anova_test* function used (within *rstatix*). Residuals were tested for normality and homogeneity of variance as described above.

*Post hoc* pairwise comparisons were done with *t*-tests and *P*-values were adjusted with Holm's method. Effect of time and community state on Phase 2 Shannon values of GROWING and STABLE were investigated using two-way repeated measures ranked ANOVA. ANOVA assumptions and model residuals were checked as described earlier. *Post hoc* pairwise comparisons were done using *t*-tests on ranked values and *P*-values were adjusted with Holm's method.

Flow cytometry data was imported using the function *fca_readfcs*[82] and analyzed using custom MATLAB scripts (v. 2021b). Flow cytometry counts of GROWING and STABLE NatCom were compared using two-way repeated measures ranked ANOVA as described above. Inoculant population sizes in conditions ALONE, GROWING, or STABLE (all time points together) were compared with a Kruskal-Wallis test followed by a *post hoc* Dunn test. The same test was used to evaluate the per-inoculant differences in average population size and fold-increase (being the maximum flow cytometry count at any time point minus the counts at start, divided by the counts at start). Differences in percentage of inoculant survival in GROWING or STABLE conditions were tested with a one-way ANOVA followed by Tukey's test. In this case one outlier per condition was removed (identified by *identify_outliers* function). ANOVA assumptions and model residuals were tested as described above. The effect of toluene on total community population size at all time points was examined with Kruskal-Wallis test (for GROWING and STABLE, and treatment, separately) followed with Dunn test (Holm's adjustment of *P*-values). Here final community sizes were also compared separately using the same approach. Changes in *P. veronii* population sizes upon introduction into toluene exposed microcosms were evaluated using Kruskal-Wallis testing with a *post hoc* Dunn test. All *P*-values from multiple pairwise comparisons were adjusted for multiple testing using Holm's *P*-value adjustment method.

The effect of inoculant or toluene on resident community compositions was evaluated by randomly pairing replicate and time point log₁₀ transformed relative abundances. First, we estimated dispersion of all paired relative abundance data (except those with toluene and inoculants removed) for either the GROWING or the STABLE microcosms in a linear regression model, and defined the outlier threshold as above or below five-fold the standard deviation of the residual distribution. Residual values were normalized by the $\log_{10}$-value taxon abundance (to correct for the $\log_{10}$-transformed data). These regression criteria were then imposed on the individual paired data sets to identify treatment-specific outliers (above = enriched, and below = depleted). The sum of the $\log_{10}$-normalized outlier residuals above or below their regression fit (relative abundance difference of each outlier to expected value from the regression) was termed the 'total outlier distance', which is reported in Fig. 3e, f. Differences in summed total outlier distances were evaluated by ANOVA as a function of microcosm treatment or evaluated by a Kruskal-Wallis test, and plotted as a function of sampling time.

Microcolony productivity distributions in agarose beads were globally compared non-parametrically with the Fisher test (implemented in R 4.0) because of their non-normal nature. Productivities of paired inoculant-soil cell taxon within the same bead were summed per grid area (e.g., as in Fig. 4c) and evaluated by comparing to *null* model simulated distributions using unpaired *t*-tests.

Paired inoculant-soil cell taxon community data were cleaned from inoculant numbers, resampled to the same total read numbers, and collapsed to Genus level, compared by pair-wise Spearman correlation and plotted as heatmap. Outlier taxa were identified at ASV level by paired comparison of absolute read abundances (corrected for biomass differences from the imaging) in triplicates in the *mattest* (implemented in MATLAB), including 999 permutations and followed by Benjamini-Hochberg correction (implemented in *mafdr*). The threshold for outliers was defined as a four-fold difference in

abundance, and a corrected $P < 0.05$. Taxon outliers were then compared between toluene-exposed bead communities and NatCom microcosms by collapsing identified individual outlier ASVs to their Families, and calculating the attributed ASVs as proportion of all the ASVs within that sample falling into the same family. Finally, bead and microcosm compositional datasets were compared in non-metric multidimensional ordination by removing inoculant reads, collapsing ASVs to Genus level, pseudonormalization by geometric means, and then calculating Bray-Curtis dissimilarities.

The main goal of the metatranscriptomic experiments with *P. veronii* in a variety of soil types was to characterize the response of the native soil community to the addition of inoculant and/or toluene. For this, all transcripts assigned to the *P. veronii* genome were removed from the metatranscriptomic data leaving only transcripts assigned to the native soil microbial community. Length-normalized transcript abundances were then calculated by dividing the total insert counts by the length of the respective gene in kilobases. Transcript abundances per kilobase (TPK) were further converted into transcripts per kilobase million (TPM) as follows[83]. The sum of TPK values in a sample was divided by $10^3$, and the result was used as a scaling factor for each sample. Each individual TPK value was divided by the respective scaling factor to produce the TPM values. Genes assigned to metabolic pathways associated to toluene and aromatic compound degradation were selected based on a pre-defined list of KEGG identifiers (Supplementary Data 6). Representative genes for some of the highly expressed pathways were taxonomically annotated by comparing to publicly available genomes. All genes from bacterial and archaeal genomes annotated to the corresponding KEGG orthologs (K15765, K16242, K00446, K07104, K04073, K10216, K05549, K16319) in IMG/M (Integrated microbial genomes and microbiomes: https://img.jgi.doe.gov/) were downloaded and used as a reference database to annotate all genes from the metatranscriptomics data with the same KEGG ortholog assignment. Global sequence alignment was performed with *vsearch* (v2.15) and genes were taxonomically assigned to the best hit (i.e., highest sequence identity; hits with a sequence identity below 70% were discarded). Genes assigned to ribosomal proteins were identified by a text-based query of the gene annotations. The relative abundance (proportion of TPM values) of all transcripts assigned to ribosomal proteins was used as an index of the native community growth rate. Indeed, levels of ribosomal protein transcripts have been shown previously to be well correlated with growth rate in yeast[84], Bacteria[85] and Archaea[86] and have been proposed and used as a metric for assessing in situ growth rates from metatranscriptomic data[87].

All statistical test results are reported in Supplementary Data 1.

### Reporting summary
Further information on research design is available in the Nature Portfolio Reporting Summary linked to this article.

## Data availability
Raw metatranscriptomic datasets of *P. veronii* inoculation into Clay, Silt, and Jonction are available from Bioproject accession number PRJNA682712, and datasets depleted from *P. veronii* reads itself can be accessed from the European Nucleotide Archive (accession numbers, ERS2210331, ERS2210332, ERS2210333, ERS2210334, ERS2210335, ERS2210336, ERS2210337,ERS2210338, ERS2210339, ERS2210340, ERS2210341, ERS2210342, ERS2210343, ERS2210344, ERS2210345, ERS2210346). The raw 16 S rRNA gene V3-V4 amplicon sequences for the random-paired inoculant-soil taxa bead communities incubated under different substrate conditions can be accessed from the Short Read Archives under BioProject ID PRJNA661487. Finally, NatCom community profiling by 16 S rRNA gene amplicon analysis is accessible through BioProject ID PRJNA1024897. All numerical data underlying the Figures, and data processing from raw data, are available as a single download from Zenodo[88].

## Code availability
All code for data processing, organised per manuscript figure, is available as a single download from Zenodo[88].

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

## Acknowledgements

The authors thank Christoph Keel for providing us with *P. protegens* CHA0 and its tagged derivative. We are very grateful for critical reading of the manuscript by Phil Gwyther. This work was supported by the Swiss National Science Foundation (Sinergia program, grant CRSII5 189919/1 to J.M.), SystemsX.ch grant 2013/158 (Design and Systems Biology of Functional Microbial Landscapes "MicroScapesX" to J.M.), and by the National Centre in Competence Research (NCCR) in Microbiomes (grant number 180575 to S.S. and J.M.).

## Author contributions
S.C., M.D., M.M., G.S. and J.M. conceived the studies and designed experiments. S.C. and V.S. performed soil microcosms. M.D. and N.C. performed microbead experiments. M.M., V.S. and N.C. conducted microcosm metatranscriptomics experiments. S.C. and J.M. analyzed community compositional and flow cytometry data; M.D. and J.M. did bead image analysis; G.S., H.-J.R., M.M. and S.S. performed metatranscriptomic bioinformatics. S.C., M.D., G.S., and J.M. analysed output data. S.C., G.S. and J.M. wrote the draft manuscript. All authors gave input, verified and corrected the written manuscript. S.S. and J.M. acquired funding and coordinated the work.

## Competing interests
The authors declare no competing interests.
