## [Peer Review File · Nature Communications]

Niche Availability and Competitive Loss by Facilitation Control
Proliferation of Bacterial Strains Intended for Soil Microbiome
InterventionsREVIEWER COMMENTS

Reviewer #1 (Remarks to the Author):

In this paper, Čaušević and co-authors performed a series of elegantly designed experiment to investigate niche competition in bacterial communities. Their study system is a soil microcosm with silt and a soil organic solution extracted from forest soil. They first described the bacterial communities in the microcosm and showed how the diversity and community composition stabilise over time. Next they inoculated either stable microcosm or growing microcosm with fresh soil extract with four species of bacteria: *Pseudomonas veronii* and *Pseudomonas putida* (which are both able to degrade toluene), *Pseudomonas protegens* and *Escherichia coli*. They showed that the first three species grow better in the batch of microcosms where nutrient were added compared to the stable microcosms while the growth *E.coli* was not different. As a third experiment, they added toluene and showed that *P. veronii* thrives when this special metabolite is added in both types of microcosms.

In the second third of the manuscript, Čaušević and co-authors performed paired assays where the focal species are encapsulated with isolated soil cells. They found that *P. veronii* grew faster alone than when encapsulated with soil cells, while the reverse was true (soil cells grow better when encapsulated with *P. veronii*), and this did not depend on the nutrient conditions.

Finally, the third part of the manuscript presents results of a metatranscriptomics experiment where *P. veronii* was grown in three different soils (silt, clay and Jonction) with and without toluene.

This paper is well-written and I enjoyed reading it. It is a very impressive piece of work which contributes greatly to our understanding of the ecology of bacterial community. I personally feel this manuscript could have been split into two separate manuscripts as there is a lot of material. I am not sure the transcriptomics part really contribute to the rest of the paper, even if I see that the authors made an effort to connect the parts of the manuscript with a hypothesis.

I have technical comments and some suggestions for the figures.

General comments

The abstract is too general and does not describe in enough details what the authors did in this paper. In my opinion, it should be entirely rewritten.

What about assumptions for ANOVA? Did you check the residuals with diagnostic plots? Please add more information in the statistical analysis subsection.

Boxplots are not appropriate to show only 4 replicates because box plot is actually a five-point summary (median + first and third quartile, and the notches). Better show the individual data point maybe with some jittering to avoid overlap. For reference: Krzywinski, M., Altman, N. Visualizing samples with box plots. *Nat Methods* 11, 119–120 (2014). <https://doi.org/10.1038/nmeth.2813>

Why do you sometimes use ASV (Figure 5) and sometimes use OTU (Figure 3)? Please pick one and use the same throughout the manuscript. I don't care which one but be consistent!

Regarding the analysis on the effect of the resident community (Figure 3C, 3D and Fig. S3). It would be interesting to see what happens if ratio normalization is used instead of calculating proportion, see: Gloor, Gregory B., and Gregor Reid. "Compositional analysis: a valid approach to analyze microbiome high-throughput sequencing data." *Canadian journal of microbiology* 62, no. 8 (2016): 692-703.

There are many established ways to identify differential abundant OTUs, for a comparison see Nearing, Jacob T., Gavin M. Douglas, Molly G. Hayes, Jocelyn MacDonald, Dhvani K. Desai, Nicole

Allward, Casey MA Jones et al. "Microbiome differential abundance methods produce different results across 38 datasets." Nature Communications 13, no. 1 (2022): 342.

I propose the authors pick one of these methods instead of the arbitrary method they used here ("differences of more than one log").

Some references are cited as superscript and others as numeral in the introduction. Pick one way and be consistent.

Line comments

line 24-26; this sentence is clumsy. To improve readability, you could add a comma after "niche to soil" and split the original sentence into two sentences. How did you improve "inoculating proliferation"? Please provide more information.

Line 34-35: the teaser is not very good. The word "suffer" is not quite appropriate in this context and the end of the sentence is not clear.

line 120. "the revived NatComs again resembled their starting material". Can you provide statistical support for this statement (for example result of PERMANOVA analysis)? Please provide statistic and associated P-value.

line 142: just before, you say that the communities differ in Shannon diversity so you cannot they that they are equivalent. Yes, their size is similar but their diversity is not.

line 162: "growth was better in GROWING than in STABLE". Can you provide statistical support for this statement? Please provide statistic and associated P-value.

There is something happening at time point 7 days and again 14 days. Most of the densities dipped, for the inoculant as well as for NatCom. Do you have an explanation for this phenomenon? This should be described in the results even if it does not affect your conclusion in the end.

Line 173: how did the authors come up with this 1% number? This is not clear to me.

Line 181: Why talk about predation since you did not test for it? Predation by whom? by other bacteria? by protist? This comes out of the blue.

line 183-184: the word toluene is repeated twice.

line 188-189: this sentence is awkward, please rephrase.

line 216: what is this total outlier distance? Please provide more information in the caption as this panel is very difficult to understand for me in its current state. What do you mean with "enriched" and "depleted" in the figure legend? Please explain.

Line 231-232: I do not understand the connection this sentence and Fig. S4.

line 250-251: I do not understand what you mean by "all the inoculants"? Do you mean *P. veronii*?

line 272-273: Why perform the analysis in this section at the family level? Please explain why you picked this level and not another taxonomic

line 273- 274: "soil extract as a sole nutrient source enabled the highest taxa diversity and growth".

Do you have some some numbers to support this statement? Maybe show in the figure the number of families which pass the threshold (>10 reads) for each treatment.

line 274: comma is missing after "As seen in the NatCom microcosms"

line 307: impact on what?

line 347: how did you come up with this number of 1%?

line 426-437: I do not quite understand the source of the soil microbiota. Could you provide more detail about how it was stored (size of the container?)

line 570: how were the simulations performed? please provide more information.

line 573: why mention a figure in the Material and Method subsection?

line 574: why mention t-test here? usually statistical analyses are collected in one subsection.

line 583 and 584: only cite figures in the material and methods if the information provided in the figure is relevant for the material and methods. In that case, explain what is shown in the figure and how does this information influence your decision. If the figures present actual results, these should be described in the result section.

line 579 (Sand and Silt) in parenthesis. What do you mean? Was one soil 100% sand and 100% silt? Please provide more description about the soils used for this experiment.

line 640-641: what do you mean "theoretical value"? These estimates are based on your data so they are no theoretical. Maybe replace with "expected values"

line 648. here you mention t-test but just before you mention Fischer test, which one did you use?

Figure 1 Panel C: why show colors for all these phyla which are not visible? It is difficult to distinguish between the different shades of purple for example. It would be best to only show phyla which are more than 1% abundant (or another threshold) to increase readability of the figure. All the phyla mentioned in the results need a color but the ones which are not visible on the barplot do not need to be shown in the figure legend.

Figure 2 Panel B. This is a very confusing figure and the legend does not describe it well. The x-axis and y-axis show counts for each media. It seems to be a scatter plot showing the relationship between the counts in the two soils, not a "difference of inoculant population size". Also the n in the corners are described as "number of dots above and below the diagonal". This is very arbitrary. How far above and below the diagonal are the dots? It would be better to use some other method of spread. There are a few timepoints when the density of the microbes dipped, for example t7 and t14, and this is visible on the scatterplots but is not described in the text. In my opinion, this panel is not bringing any useful information.

Figure 3 Panel A and B: please use the same color scheme for both panels, so the STABLE subpanel A should have lighter shades of purple and green as in panel B.

The pie chart and barplot from Figure 3 and Fig. S3 have too many colors and it is not possible to see what is happening. I suggest to use a threshold and only show 10 or 15 taxa above the threshold which can be seen on the plots and which are also described in the text. In addition, provide supplementary tables which list all the taxa which are unregulated or down regulated upon addition of toluene.

Fig. S4: why show colors for all these phyla which are not visible? see comment above. Please only show 10 or 14 taxa, it is not possible with the eyes to see the difference between so many shades.

Fig. S4 top of panel. what is the different between the first facet ("mixed C") and the second facet (also "mixed C"). The text at the bottom also seems identical (T0, T48, T48, -, -, Pve).

Figure 4: It is confusing to have this abbreviation "SC" in the title, there is enough space to write these words in full. Do not introduce unnecessary abbreviations.

Figure 5 Panel A: It is not clear to me if a selection of families are shown in this panel or if all the detected families are shown. What are the numbers on the x-axis?

Figure 5: I don't understand the legend with the different colors at the family level. Why have B Burholderiaceae and C Caulobacteriaceae but no color for those two families? If those families are absent in the figure, please drop them in the legend. There are some colors in Panel C which are not described in the figure legend, for example orange. Upon reading the legend for the third time, I see that panel C shows ASVs but I find colors are not helpful here and this whole panel is very confusing. Best would be to present these results as supplementary table or drop the panel entirely.

Reviewer #2 (Remarks to the Author):

This work investigates the importance of nutritional niche availability in allowing introduced bacterial species to thrive within resident soil microbial communities. By comparing introduction of various *Pseudomonads* and *E. coli* to stable versus growing natural communities, the authors find that introduced species proliferate more when co-introduced with a growing soil community and when inoculated into soil amended with an inoculum preferred carbon source like toluene.

The authors also demonstrate through agarose bead encapsulation experiments that the inoculants benefit the growth of resident soil community taxa and that conversely, the presence of different soil community taxa negatively impact growth of inoculant microbes. The authors describe this phenomenon as competitive loss by facilitation whereby inoculant microbe metabolism generates additional metabolites and by-products that soil microbes can utilize. The authors go on to further show that the competitive loss by facilitation is also evident within non-sterile soils that are more representative of in situ conditions. Here they use metatranscriptomics to elucidate abundance of resident communities in response to *P. veronii* inoculation and observe there is an increase in their abundance in the presence of *P. veronii* for some soil types. They also find the resident community has an enrichment for pathways in toluene by-product degradation pathways, indicating a likelihood for cross-feeding between *P. veronii* and the resident microbial community.

The authors did a thorough investigation of their hypothesis of competitive loss by facilitation however the novelty of these findings could be more clearly communicated. It is not evident from the way the manuscript is written whether this is a newly described phenomenon or if it is confirming previous studies that have also shown this trend. Given that the manuscript title includes "competitive loss by facilitation", it would be helpful for the authors to discuss whether this phenomenon has been observed previously or whether their study is the first to demonstrate it.

This study uses robust methodology to demonstrate the importance of nutritional niche availability on microbiome engineering, however it does not investigate other factors impacting microbiome engineering like presence of predator microorganisms, environmental tolerance (to abiotic factors in the soil like salinity, pH, etc), effect of inoculum size or dispersal mechanism. Given these other factors are not investigated in this work, the conclusions made should also clearly state they are

limited to nutritional niche availability and that other factors may also influence some of the observed results.

Comments that require revision before publication:

Line 20: the authors should list the ecological factors they study in this work.

Line 24. The way to describe addition of toluene as "inoculant selective niche" seems like an overstatement, especially considering the growing NatCom also appeared to have improved growth in response to toluene. I would recommend rewording it to "inoculant preferred carbon source".

Line 30. The term "selective niche generation" here is not clear. The addition of toluene greatly influenced the diversity of the NatCom (Figure S3) without the presence of any introduced *P. veronii* as well.

Line 54. Is this accurate there is no previous work that aids in resolving why newly inoculated strains don't establish in target microbiomes? It would be useful to provide some references here to support this otherwise reword "remain unresolved" to "remain largely unresolved".

Line 56. Is this the correct reference for this sentence? Reference 16 does not describe niche availability as a factor for determining successful inoculant proliferation. Please replace with the correct citation to support this statement.

Line 59-60. Is this the correct reference for this statement? The reference cited describes how transiently invading microbes have a permanent impact on the resident community and does not appear to describe how emerging function of resident microbiome disfavours opportunities for new strains to grow.

Line 72. This work doesn't discuss or demonstrate through experimentation how the introduced inoculum provides plant-growth beneficial functions. It would be good to see more discussion around this.

Line 77. It would be good to describe the NatCom in more detail here to give context on the abiotic and biotic features of the soil (ie. Presence of predator species? pH, salinity? Available carbon sources?).

Line 90. When the term "niche availability" is used throughout, it should be more specifically "nutrient niche availability" as this was the only aspect investigated (ie. Through addition of toluene). If the authors also tested different dispersal introduction methods this would examine spatial niche availability and also if they tested different pH gradients this would examine abiotic niche availability.

Line 81 and 91. The term inoculant-specific niche is broad as it seems to be just a nutritional niche that is created. I would recommend changing to "inoculant-specific nutritional niche" or "inoculant preferred carbon source".

Line 109. It would be helpful to describe the carbon source profile of the soil organic carbon solution and whether the introduced *Pseudomonas* sp. and *E. coli* have the metabolic capacity to utilize the available carbon sources.

Line 120. Was the transition of the NatCom back to its original state within a month expected? It would be helpful to include a comment on whether this is expected given what is known from previous studies.

Line 125. Were the microcosms split into 45 before phase 1 for a particular reason? Since they were

pooled at the end of Phase 1, it is not clear why they were split in the first place.

Line 133. Was it expected the growing NatCom remained lower than stable NatComs during most of phase 2? Is there some level of re-adaptation to the fresh soil microcosm and soil extract that limits their ability to reach as high titres as stable? A sentence or two around whether this was expected would be helpful.

Line 148-149. The inoculum size of freshly diluted growing NatCom versus stable NatCom at $t=0$ in phase 2 is different ($\sim 5 \times 10^7$ for stable vs $\sim 5 \times 10^6$ for growing as shown in Fig 1E). This 10-fold difference in inoculum size of NatCom is an additional factor to nutrient replete vs deplete conditions that would affect niche availability for the introduced *Pseudomonas* sp and *E. coli* which were inoculated at a lower inoculum (1×10^5). Hence, the niche availability in growing vs stable is higher both spatially and nutritionally and this should be clearly stated.

Line 173. Since no analysis of the nutrient profile was done to confirm this and also since the growing NatCom condition would also have more spatial niche available, it is more accurate to say the "potential nutrient and spatial niche". Also, how is the 1% of nutrient niche available calculated? Is this based on growth of the introduced species? There should be a reference to the methodology for calculating this as its' not obvious how this percentage was obtained.

Line 181. What does a priori predation mean? Was predation measured in this work?

Line 189-190. Was it expected that the growing NatCom would have a higher ability to utilise toluene than the stable NatCom? A sentence here about whether this was expected and why would be helpful.

Line 195. Has this been reported as well in other studies? A comment about whether this is a novel finding is important to include here.

Line 217-219. The result that toluene had a greater impact on NatCom diversity in the absence of *P. veronii* compared to the presence of *P. veronii* is not explained. This seems important as it shows the effect of toluene greatly shifts NatCom diversity and this effect is dampened by the presence of *P. veronii*. Was this expected? What is the reason this occurs? More discussion around this significant trend is warranted.

Fig S3 A) is missing 2 graphs that it mentions in the caption are included, the without toluene (and no *P. veronii*) impact on growing and stable NatCom diversity.

Line 221. Can the authors elaborate on how *P. veronii* inoculation alleviated negative effects of toluene exposure on NatComs? This is not clearly explained. It is because there were more pronounced taxa changes for NatComs exposure to toluene without *P. veronii* inoculation? Since Fig S3 is missing two graphs of no toluene or *P. veronii* exposure, it is not clear if there are also significant taxa changes in this condition.

Line 225. This section needs to be introduced more clearly in terms of how the experiment was designed. There is no rationale given for why sand extract was used and why (and what) 16 carbon sources were used for the Mixed C condition.

Lines 228-234. I found this section difficult to understand without referring to the methods section. It would be helpful for the authors to make it clear that each bead contained on average 1 inoculant cell plus 1 soil NatCom species. Also, clearly state that "founder cell pairs" refers to the ~ 2 initial cells inoculated into each bead.

Line 239. Do the authors mean "sand" rather than "soil" extract here? According to Fig. 4A, it is sand extract.

Line 245-246. It would be helpful to have a number stated here for average size of SC Alone vs SC with PV as it is hard to see from Fig. 4A how significant the difference in size is. Also, the authors state that SC alone in mix shows similar average growth to SC alone, however the p-values shown in Fig. 4A suggest there is a significant difference in average growth between SC alone and SC alone in mix.

Line 273. Soil extract is mentioned in text but in Fig. 5 it is described as sand extract. Also same issue in Line 284.

Line 312. What is the possible reason that clay soils did not allow for as much *P. veronii* growth? Were resident microbiota transcripts in clay associated with aromatic compound metabolism and they just outcompeted *P. veronii* for toluene utilization?

Line 316. Did the transcript of silt resident communities become enriched in aromatic compound metabolism later in the incubation? It appears this is the case in Fig 6D. A sentence should be added here to describe this observation as it offers an explanation of why their higher abundance of ribosomal protein transcripts occurred later in incubation as they adapted to toluene utilization. It is also useful for the authors to mention the observation that in the presence of *P. veronii* in silt conditions at later time points, the transcript levels for toluene degradation pathway genes were lower than that for those in the absence of *P. veronii*. This would suggest the resident soil community adapts more strongly to toluene utilisation pathways in the absence of *P. veronii* which makes it grow slightly less (as seen in Fig. 6B).

Line 320. Fig 6D does not show "inoculated toluene-free". I think the authors meant to say "uninoculated toluene-free" here.

Line 324. Fig S6 and S7 do not show the transcript levels for Junction + toluene but uninoculated with *P. veronii*. This figure would be helpful to include to support the statement that is made in this sentence.

Line 332. I believe the authors mean Fig. S8 not S7 here.

Line 342. I would recommend rewording to "available nutritional niches" as those are the particular niches that were investigated.

Line 347. As mentioned earlier, it would be useful to know how is the 1% calculated?

Line 351-355. It is difficult to understand what the authors are trying to say here. I recommend this sentence is reworded and made clearer. Also, given that the initial inoculum size of the growing community is higher than that of the *Pseudomonas* inoculants, the fact that more spatial niche will be occupied initially by the growing community is also worth mentioning.

Line 356. Not sure if the term "leakage of metabolites" is accurate here, consider rewording or explain what this means more clearly. For instance, does it refer to the fact that some of the resident soil community is able to utilise the toluene on its own and thus they are able to use this as a carbon source in addition to the by-products of toluene degradation generated by *Pseudomonas* toluene metabolism?

Line 361. The concept of growth inhibition of the inoculant warrants further explanation given it is mentioned for the first time in the manuscript here. How does it differ from competitive loss? Was there any investigation into growth inhibiting factors?

Line 368-369. Is this stated expectation based on previous literature or a hypothesis? Include citations if based on previous literature. Also, the conclusions made in line 374-375 do not correspond with the

stated expectation that growing communities would limit proliferation of incoming species since in line 373, the authors state the observed result that the inoculants established better in growing than stable NatCom conditions. This section needs to be re-worded to improve clarity.

Line 378. The authors should be more specific they are observing niche "nutrient" (un)availability and competition "for nutrients".

Line 382. Is this the first study to show this is applicable to soil microbiota interventions? If no then include citations of other studies that show this. If yes, then state it explicitly as it highlights the impact of this work.

Line 385. "potential biological interactions" sounds too vague. I would suggest re-wording this to "potential interspecies cross-feeding interactions".

Line 390. "facilitating the growth of other soil bacteria" is also too vague. I would recommend the authors be more specific what mode of facilitation they identified in their work (ie. Cross-feeding).

Line 400-402. The concept of temporal available niches is brought up for the first time here and should be defined. It's not clear how the results of this study demonstrates establishment of temporal niches. This discussion point seems out of place and should be clarified.

Line 405. I would specify "nutritional niche engineering" as this study mainly focused on investigating nutritional niches.

Line 409. The concluding sentences need some additional work. For instance, this study showed that bioremediation agents would not exactly "thrive thanks to the selective niche provided" but instead be restricted through competitive loss by facilitation. I think the authors need a stronger conclusion on how their work can inform microbiome engineering for bioremediation purposes.

Line 441. Can the authors describe the expected carbon sources present in the forest soil extract? This would help give context as to why *E. coli* had poor growth in the soil microcosms.

Fig 1 A). make bottle grey to not confuse with the different shades of blue to describe NatCom in soil (18 mo and 1 mo). Also make fresh soil + extract vs 1 mo old soil + extract Phase 1.

Fig S3 B), Fig S6 and Fig S8 - the font in these figures is too small.

Reviewer #3 (Remarks to the Author):

Review of "Niche Availability and Competitive Loss by Facilitation Control Proliferation of Bacterial Strains Intended for Soil Microbiome Interventions" by Čaušević et al.

Manuscript ID: NCOMMS-23-49794

In the paper titled "Niche Availability and Competitive Loss by Facilitation Control Proliferation of Bacterial Strains Intended for Soil Microbiome Interventions," Senka Čaušević et al. explore the complex dynamics of introducing bacterial inoculants into soil communities. The study primarily focuses on understanding the proliferation of typical soil inoculants, such as *Pseudomonas* spp., in

relation to niche availability and competition within diverse soil microbial communities (NatComs). The research employs a combination of methodologies, including the cultivation of standardized soil microbiomes, randomized paired growth assays, and metatranscriptomic analysis. The findings reveal that the proliferation of soil inoculants is influenced by limited niche availability and their competitive disadvantage arising from facilitating the growth of rival populations. This situation can be altered by creating selective niches, such as introducing bioavailable toluene, which enhances inoculant survival and integration into the soil microbiome. This work underscores the significance of niche dynamics in microbial community interventions and suggests that strategic niche manipulation can improve the successful integration of inoculants into soil ecosystems. The results offer crucial insights for microbiome engineering, particularly in the realms of soil health and bioremediation strategies.

While the paper is well-articulated and offers a novel perspective on the interactions of bacterial inoculants within soil microbiomes, it would greatly benefit from a deeper re-analysis of the data to provide a more comprehensive understanding of the results. This includes a more detailed and nuanced description of how the inoculants affect community composition, as well as a robust comparative analysis of the inoculant's impact on the entire community versus its effect in random pairwise experiments. Such an analysis could reveal intricate patterns and dynamics that are not immediately apparent, thereby enriching the study's conclusions. Addressing these aspects is crucial for a more complete and accurate representation of the microbial interactions and their ecological implications. Thus, I recommend publication, provided that the authors undertake this comprehensive data re-analysis and enhance the discussion to more effectively contextualize and interpret the study's findings within the broader scope of microbial ecology and soil microbiome research.

Comments on the Abstract

The last sentence of the Abstract contains the phrase "competitive loss by facilitation," which may lead to ambiguity. In ecological contexts, "facilitation" typically denotes a positive interaction benefitting at least one participant, usually without notable detriment to the facilitator. Therefore, the expression "competitive loss by facilitation" could be misinterpreted as it implies a reduction in competitive ability through a mechanism generally viewed as advantageous.

Maybe this can be solved by something like: "Thus, the fate of inoculants in soil is governed by niche availability and their competitive disadvantage resulting from facilitating rival populations, a dynamic that can be manipulated by selective niche generation."

Comments on the Introduction:

1. In the introduction, lines 56 to 63 explore reasons why inoculant bacteria might struggle to establish themselves in soil microbial communities, focusing on competition for resources and habitat changes that impede their survival or interaction with resident microbes. The subsequent lines, 63 to 65, point out a gap: these theories, particularly regarding resource competition and microbial interactions, haven't been rigorously tested in experiments involving both the microbiome and inoculants. Moreover, the text notes that existing mechanistic theories are largely based on a narrow range of strains, often pathogenic. This raises a critical question: What exactly are these "mechanistic concepts," and how do they connect to the ecological processes discussed earlier? Are these concepts applicable to the broader dynamics of soil microbial communities, or are they too limited in scope?

2. Similarly, the reference to "underlying mechanisms" in lines 65-68 remains vague. The text

mentions "N+1/N-1 engineering," a concept purportedly useful for revealing mechanisms of community assembly and development, yet fails to elucidate what this concept entails. This omission leaves a gap in the introduction's conceptual framework, raising more questions than it answers. Additionally, the current wording does not clarify whether N+1/N-1 engineering applies solely to pathogenic strains or has broader applicability. For an introduction to effectively set the stage for the results, it should clearly articulate these underlying mechanisms, especially those influencing the success or failure of inoculant strains in various ecological contexts. Simply stating the existence of a mechanism or limiting discussion to pathogenic strains, without adequately explaining the introduced concept, falls short of providing the necessary context for the study's findings.

General Comment for the Results section:

In reviewing the manuscript, I suggest that the authors carefully reconsider the placement of certain details within the Results section. Specifically, information such as, "All inoculants constitutively expressed an introduced genetically encoded mCherry tag, which facilitated the quantification of their specific population size," appears more appropriate for the Materials and Methods section. The inclusion of such methodological details within the Results section disrupts the narrative flow and impedes a clear and concise presentation of the findings. Relocating these details would enhance the readability and coherence of the Results section.

Additionally, it is imperative that the authors refine the logical progression of the Results section. The current structure presents the findings as a series of isolated observations rather than a cohesive narrative. A more effective approach would be to ensure that each segment within the Results logically connects to the next. This will facilitate a comprehensive understanding of the research, allowing readers to follow the progression of the study and appreciate how each result builds upon the previous ones, ultimately contributing to a unified understanding of the underlying processes. Such a structured approach is crucial for effectively communicating the significance and implications of the research findings.

1. In Lines 136-138, the authors state, "taxa richness was initially lower than that of STABLE NatComs, but became similar from Day 14 onwards (Fig. 1F). This apparent lower diversity..." In this context, there appears to be a conflation between 'taxa richness' and 'diversity.' Taxa richness pertains solely to the number of species present, whereas diversity is a more comprehensive metric that includes both the number of species and their relative population sizes. The distinction between these two concepts is critical in ecological studies, yet the authors seem to use them interchangeably.

The manuscript predominantly features taxa richness (as depicted in Fig. 1F) while relegating the analysis of diversity to the supplemental material (Fig. S1). This decision is not explicitly rationalized in the text. The criticality of this choice becomes evident in the concluding sentence of this section: "We thus concluded that, while the dynamic succession of GROWING and STABLE NatComs varied, the equivalent richness and size of either community meant they were suitable for testing the effect on inoculant proliferation." This conclusion implies that having equivalent taxa richness is a necessary condition for the GROWING and STABLE NatComs to be considered appropriate for the inoculant proliferation experiments. However, this inference potentially overlooks the importance of diversity in determining the ecological dynamics and experimental outcomes.

Furthermore, there is an issue with the clarity and precision in the reporting of statistical results. On Day 56 in Fig. S1, the p-value is ambiguously reported as "p=00618," leaving it unclear whether this should be interpreted as p=0.00618 or p=0.0618. This distinction is crucial, as it determines the statistical significance of the diversity measures. In a scientific context where precise and unambiguous reporting of statistical data is paramount, this lack of clarity could lead to misinterpretations, casting doubt on the validity of the study's conclusions.

In summary, the authors' treatment of taxa richness and diversity raises concerns regarding both the ecological interpretation and the statistical robustness of their findings.

-

2. In addition to the previously discussed issues regarding the interpretation and reporting of taxa richness and diversity, a further exploration into the study's approach to community composition metrics is necessary. The authors have presented measures of abundances (FCM counts per gram of soil) and richness (Chao1 index value) in Fig. 1E and Fig. 1F, as well as diversity (Shannon index value in Fig. S1). These metrics, while informative, may not fully capture the complexities of community dynamics.

A key question arises in this context: Can one infer whether a sample belongs to the GROWING or STABLE scenario based on its specific relative abundance profile? Addressing this question would significantly enhance our understanding of the community dynamics at play. The current methodology, focusing primarily on simpler metrics like richness and abundance, may overlook the intricate patterns that emerge from more detailed analyses of relative abundance profiles.

To address this gap, it would be advantageous for the authors to consider applying advanced analytical methods, such as K-means clustering analysis. Such an approach could elucidate whether distinct relative abundance profiles are characteristic of the GROWING and STABLE community states, and whether these states form separate clusters. This analysis would not only provide a deeper insight into the community structures but also add robustness to the conclusions drawn about the dynamics and physiological states of the communities under study.

3. Lines 210-211: "In contrast, more dramatic shifts in taxa abundances were observed in presence of toluene and, perhaps counterintuitively, more in STABLE than GROWING NatCom (Fig. 3D)." Could the authors elucidate the reasons behind these significant shifts in taxa abundances?

4. The initial sections of the results focus on the impact of the inoculum on complex soil communities, while the fourth section adopts a more reductionist yet quantitatively precise approach, examining effects through random pairwise trials. This distinction is noteworthy and praiseworthy for its comprehensive analysis. An intriguing question arises from this: can we link the growth patterns observed in these trials to those seen when inoculants interact with the full community? Specifically, is there a way to correlate species' growth rankings from pairwise trials with their behavior in the context of the entire community? Such analysis would be valuable in quantifying the extent and intensity of nonlinear interactions that occur in a diverse community setting. This comparison would enhance our understanding of how inoculants interact differently in isolated pairwise setups versus more complex, multi-member community environments.

Comments on the Discussion section

The discussion section tends to revisit the results in excessive detail, which could be streamlined for brevity. A more effective approach would be to succinctly recap the key findings and then focus on integrating these results within the context of existing literature. Each time the results are referenced, they should be directly connected to relevant studies, thus situating the paper's contributions within the broader scientific discourse. This approach would not only enhance the clarity of the discussion but also underscore the significance and novelty of the study's findings in advancing our understanding of the field.

REVIEWER COMMENTS

Note : all line numbers based on the document '...all_changes_marked.pdf'.

Reviewer #1 (Remarks to the Author):

In this paper, Čaušević and co-authors performed a series of elegantly designed experiment to investigate niche competition in bacterial communities. Their study system is a soil microcosm with silt and a soil organic solution extracted from forest soil. They first described the bacterial communities in the microcosm and showed how the diversity and community composition stabilise over time. Next they inoculated either stable microcosm or growing microcosm with fresh soil extract with four species of bacteria: *Pseudomonas veronii* and *Pseudomonas putida* (which are both able to degrade toluene), *Pseudomonas protegens* and *Escherichia coli*. They showed that the first three species grow better in the batch of microcosms where nutrient were added compared to the stable microcosms while the growth *E.coli* was not different. As a third experiment, they added toluene and showed that *P. veronii* thrives when this special metabolite is added in both types of microcosms.

In the second third of the manuscript, Čaušević and co-authors performed paired assays where the focal species are encapsulated with isolated soil cells. They found that *P. veronii* grew faster alone than when encapsulated with soil cells, while the reverse was true (soil cells grow better when encapsulated with *P. veronii*), and this did not depend on the nutrient conditions.

Finally, the third part of the manuscript presents results of a metatranscriptomics experiment where *P. veronii* was grown in three different soils (silt, clay and Jonction) with and without toluene.

This paper is well-written and I enjoyed reading it. It is a very impressive piece of work which contributes greatly to our understanding of the ecology of bacterial community. I personally feel this manuscript could have been split into two separate manuscripts as there is a lot of material. I am not sure the transcriptomics part really contribute to the rest of the paper, even if I see that the authors made an effort to connect the parts of the manuscript with a hypothesis.

We thank the reviewer for the overall appreciation of our work. We prefer keeping the material together, since it carries a coherent message on the fate of soil inoculants and the role of nutrient niches.

I have technical comments and some suggestions for the figures.

General comments

1) The abstract is too general and does not describe in enough details what the authors did in this paper. In my opinion, it should be entirely rewritten.

Reply and action: We thank the reviewer for this statement. We have rewritten the abstract and have added more details.

2) What about assumptions for ANOVA? Did you check the residuals with diagnostic plots?

Please add more information in the statistical analysis subsection.

Reply and action: Yes, ANOVA assumptions were checked (see e.g., l. 1168-1177). All details on statistical procedures had been included in the S1 dataset. We have updated this supplementary document according to the new analyses included in the revised manuscript.

3) Boxplots are not appropriate to show only 4 replicates because box plot is actually a five-point summary (median + first and third quartile, and the notches). Better show the individual data point maybe with some jittering to avoid overlap. For reference: Krzywinski, M., Altman, N. Visualizing samples with box plots. *Nat Methods* 11, 119–120 (2014). <https://doi.org/10.1038/nmeth.2813>

Reply and action: We thank the reviewer for reminding us of this. Our reason for using the box plots was also to have some 'anchoring' of the individual points in the diagram 'space'. However, we have corrected this in all figures where this was relevant to showing only the individual data points and their mean.

4) Why do you sometimes use ASV (Figure 5) and sometimes use OTU (Figure 3)? Please pick one and use the same throughout the manuscript. I don't care which one but be consistent!

Reply: We thank the reviewer for the remark. In Figure 5, we aimed to target a lower taxonomic level to describe potential variations in bead incubations. We realize that this has created confusion.

Action: We revisited the analysis of Figure 5 to display ASV enrichments from the bead diversity data of the sand extract and toluene incubations, which are the most relevant to compare to the soil microcosms (panel B and a new supplementary figure S5). These data are now also corrected for the absolute biomass differences in the samples from the image analysis. To compare the enrichments among the five bead incubations, we collapsed the ASVs to genus level and removed the inoculants themselves, to better detect any changes in the background community (this is now panel A). We then added a new analysis – also based on remarks by reviewer 3, to compare the bead enrichments with the microcosm enrichments as a consequence of inoculation or toluene treatment. Here we collapse the enriched ASVs from both systems to their Family level, and display the proportion of increased and decreased ASVs with their relative read abundances (panel C). This, we feel, better highlights whether differences occur consistently across the soil microcosms and the bead communities. Finally, we compare the soil microcosm and bead communities as a whole, as a function of inoculant and treatment, using non-metric multidimensional scaling analysis (on pseudonormalized read abundances). This is the new panel D and shows more clearly the effects of treatment and (common) inoculant.

5) Regarding the analysis on the effect of the resident community (Figure 3C, 3D and Fig. S3). It would be interesting to see what happens if ratio normalization is used instead of calculating proportion, see:

Gloor, Gregory B., and Gregor Reid. "Compositional analysis: a valid approach to analyze microbiome high-throughput sequencing data." *Canadian journal of microbiology* 62, no. 8 (2016): 692-703.

There are many established ways to identify differential abundant OTUs, for a comparison see Nearing, Jacob T., Gavin M. Douglas, Molly G. Hayes, Jocelyn MacDonald, Dhvani K. Desai,

Nicole Allward, Casey MA Jones et al. "Microbiome differential abundance methods produce different results across 38 datasets." Nature Communications 13, no. 1 (2022): 342.

I propose the authors pick one of these methods instead of the arbitrary method they used here ("differences of more than one log").

Reply: We thank the reviewer for these suggestions and we apologize for not having been more clear.

Action: We have explored two different methods to detect and quantify changes in relative abundances, which largely converge to the same ASVs in the various samples. In our preferred method, we use linear regression on the paired log₁₀-relative abundance data across all time points of all inoculations (except those with toluene, which are too 'disturbed') to define the general dispersion of the residuals, and then use 5 times the standard deviation of the residual distribution to mark the outlier range. This is what is now shown in the revised Figure 3 panels C and D.

In the second method, we applied paired triplicate tests as is typical for gene expression data, with multiple test correction, to identify differences with an adjusted p-value of <0.05 and a fold-change of >2. This, however, is less practical for the comparison across multiple time points and across multiple treatments.

The lists of defined outlier ASVs is presented in the new supplementary Tables S1 and S2.

*In particular the outlier taxa in the microcosms exposed to toluene and inoculated with *P. veronii* are then compared to those from the paired bead studies in a new Figure 5C.*

6) Some references are cited as superscript and others as numeral in the introduction. Pick one way and be consistent.

Reply and action: We apologize if this occurred. The Endnote formatting should be consistent here but it evidently wasn't. We verified this in the revised manuscript.

Line comments

7) line 24-26; this sentence is clumsy. To improve readability, you could add a comma after "niche to soil" and split the original sentence into two sentences. How did you improve "inoculating proliferation"? Please provide more information.

Reply and action: we apologize if the sentence looked clumsy. We added a specification that "inoculant proliferation" was improved by engineering a specific nutrient niche in form of toluene addition. The sentence was split in two (and anyway, the whole abstract was reworked to be more informative).

8) Line 34-35: the teaser is not very good. The word "suffer" is not quite appropriate in this context and the end of the sentence is not clear.

Reply and action: This was changed to: "fail to establish because they inadvertently facilitate..."

9) line 120. "the revived NatComs again resembled their starting material". Can you provide

statistical support for this statement (for example result of PERMANOVA analysis)? Please provide statistic and associated P-value.

Reply: All the statistical values and results were provided in the S1 data table. The notion of resemblance was based both on the PERMANOVA comparison of Day 23 to Day 0 ($p=0.054$) and the time trend shown in the Fig 1D plot.

Action: We revisited this analysis and the accompanying statements in l. 212-217, describing the Chao1 and Shannon index comparisons, the PCA on Unifrac distances and the PERMANOVA following the ordination.

10) line 142: just before, you say that the communities differ in Shannon diversity so you cannot they that they are equivalent. Yes, their size is similar but their diversity is not.

Reply: We thank the reviewer for the remark. To our defense, we wrote that richness and size were equivalent, not 'diversity' as a whole.

Action: See our reply to point 9. This section (l. 212-216) was rewritten.

11) line 162: "growth was better in GROWING than in STABLE". Can you provide statistical support for this statement? Please provide statistic and associated P-value.

Reply : We apologize that for readability we did not again specify all test results in the main text. All statistics and p-values were reported in the figure panels with all details in S1 data, and in this case the associated p-values were reported in the figure panel itself with mention of the test in the figure legend.

Action: We specify here and elsewhere when test results are in the figure (e.g., l. 263).

12) There is something happening at time point 7 days and again 14 days. Most of the densities dipped, for the inoculant as well as for NatCom. Do you have an explanation for this phenomenon? This should be described in the results even if it does not affect your conclusion in the end.

Reply and action: We think this has no biological meaning but is the unfortunate consequence of the difficulty of the flow cytometry gating in presence of soil particle background that in some samples led to an underestimation of the actual cell counts. For this reason we group measurements over different time points when making statements about treatment effects. We explain and show flow cytometry gating procedures in a new supplementary Figure S9.

13) Line 173: how did the authors come up with this 1% number? This is not clear to me.

Reply: We apologize if this wasn't clear. As written, it was calculated from the "difference in inoculant proliferation in axenic microcosms compared with" STABLE or GROWING communities.

Action: We specified the exact values of the mean ratios over all time points (l. 352-355).

14) Line 181: Why talk about predation since you did not test for it? Predation by whom? by other bacteria? by protist? This comes out of the blue.

Reply: We mention predation because this has been often proposed as a factor in controlling inoculant proliferation (see, e.g., reference 22 and 23), but we agree that mentioning this here came a bit out of the blue.

Action: Also in the light of remarks of reviewer 2, we removed this part of the sentence here (l. 363).

15) line 183-184: the word toluene is repeated twice.

Reply: Thank you. Was deleted.

16) line 188-189: this sentence is awkward, please rephrase.

Reply: We reworded this sentence (l.370-372).

17) line 216: what is this total outlier distance? Please provide more information in the caption as this panel is very difficult to understand for me in its current state. What do you mean with "enriched" and "depleted" in the figure legend? Please explain.

Reply: The total outlier distance is the sum of the residuals of the identified outlier taxa – those with relative abundance differences to the linear regression on all data higher than 5 times the standard deviation of the residuals in linear regression. Enriched taxa are then those with higher than expected relative abundances in presence of the inoculant or treatment; depleted the opposite.

Action: We have explained and reworded this text on l. 432-434 and exported the relevant taxa in Table S1. The busy Figure 3C and D panels were simplified using taxa abbreviations. The corresponding methods and statistics is described in the revised text in l. 1215-1226.

18) Line 231-232: I do not understand the connection this sentence and Fig. S4.

Reply : We apologize for the confusion. Our intention was to show the taxa diversity coming from the soils used for the pairs-in-bead experiments.

Action: Given that the reviewer asked for a new analysis at ASV level, we removed Fig. S4 and replace this by a new Fig. 5 and associated supplementary data graph Fig. S5, and new Tables S3 and S4 with the diversity data on the beads and the identified outlier taxa. The new comparison added between soil microcosm and bead community outliers is described in lines 1215-1226.

19) line 250-251: I do not understand what you mean by "all the inoculants"? Do you mean P. veronii?

Reply: The "all (...) inoculants" here refers to the four inoculants of panel 4B.

Action: The procedure was re-explained and the paragraph was rewritten to make this clearer (l. 520-525).

20) line 272-273: Why perform the analysis in this section at the family level? Please explain why you picked this level and not another taxonomic

Reply: Our aim was to be able to compare the bead data with the microcosm data, in order to see if the types of bacteria enriched or depleted as a function of substrate or inoculant would be

similar. Because the soil used as community starter for the bead experiments was not the same as the NatCom microcosms, their ASVs would not be the same and we would need a higher taxonomic level for comparison.

Action: This is explained in our reply to the general remark above. In essence, Figure 5 was revised entirely – we use ASV levels and higher order taxonomy comparison, and rewrote this paragraph (l. 626-633).

21) line 273- 274: "soil extract as a sole nutrient source enabled the highest taxa diversity and growth". Do you have some some numbers to support this statement? Maybe show in the figure the number of families which pass the threshold (>10 reads) for each treatment.

Reply: We thank the reviewer for the suggestions. The ASV richness in the incubations with sand extract is on average twofold higher.

Action: We added the richness numbers to Figure 5 and the ANOVA statistics to the S1 Data (l. 626).

22) line 274: comma is missing after "As seen in the NatCom microcosms"

Reply: Thank you. This paragraph was rewritten based on the revised Figure 5.

23) line 307: impact on what?

Reply: Thank you. We specified "impacts on the soil resident community" (l. 709)

24) line 347: how did you come up with this number of 1%?

Reply: This was explained in our reply to remark 13 above and changed accordingly (l. 753).

25) line 426-437: I do not quite understand the source of the soil microbiota. Could you provide more detail about how it was stored (size of the container?)

Reply : Also in light of suggestions by reviewer 2, we added more information to this paragraph, and specified the size of the bottles and the stored soil volume (l. 936-939).

26) line 570: how were the simulations performed? please provide more information.

Reply and action: We explain this in more detail in l. 1090-1092/1103-1105), and provide the actual script code from the open data source on Zenodo. (Ref. 88)

27) line 573: why mention a figure in the Material and Method subsection?

Reply: We mention the figure here to make the procedure of the simulation more clear.

Action: No further action.

28) line 574: why mention t-test here? usually statistical analyses are collected in one subsection.

Reply and action: We removed the test description here as suggested, to place it in the appropriate subsection (l. 1154 etc.).

29) line 583 and 584: only cite figures in the material and methods if the information provided in the figure is relevant for the material and methods. In that case, explain what is shown in the figure and how does this information influence your decision. If the figures present actual results, these should be described in the result section.

Reply and action: This was removed here. There was no specific 'results' information within the materials and methods.

30) line 579 (Sand and Silt) in parenthesis. What do you mean? Was one soil 100% sand and 100% silt? Please provide more description about the soils used for this experiment.

Reply and action: We specify the source material of both soils and Jonction (l. 1111).

31) line 640-641: what do you mean "theoretical value"? These estimates are based on your data so they are no theoretical. Maybe replace with "expected values"

Reply and action: This paragraph (l. 1215-1226) was rewritten to better explain the (changed) procedure of regression and outlier definition.

32) line 648. here you mention t-test but just before you mention Fischer test, which one did you use?

Reply: We apologize. These are two different data set comparisons: histograms of productivities among treatment (e.g., as in Figure 4A and B), and comparisons of PAIRED productivities between observations and null model simulations (as in Figure 4C). We explained this better in line 1228-1231.

33) Figure 1 Panel C: why show colors for all these phyla which are not visible? It is difficult to distinguish between the different shades of purple for example. It would be best to only show phyla which are more than 1% abundant (or another threshold) to increase readability of the figure. All the phyla mentioned in the results need a color but the ones which are not visible on the barplot do not need to be shown in the figure legend.

Reply and action: Thank you. We followed your suggestion and now only show colors of the most abundant taxa (Revised Figure 1C).

34) Figure 2 Panel B. This is a very confusing figure and the legend does not describe it well. The x-axis and y-axis show counts for each media. It seems to be a scatter plot showing the relationship between the counts in the two soils, not a "difference of inoculant population size". Also the n in the corners are described as "number of dots above and below the diagonal". This is very arbitrary. How far above and below the diagonal are the dots? It would be better to use some other method of spread. There are a few timepoints when the density of the microbes dipped, for example t7 and t14, and this is visible on the scatterplots but is not described in the text. In my opinion, this panel is not bringing any useful information.

Reply : We apologize if this figure panel caused confusion. It was meant to show more clearly if there would be a significant trend of an inoculant proliferating better under GROWING or under STABLE condition, but in a way this is already shown in panels C-E.

Action: We removed this panel from the revised Figure 2.

35) Figure 3 Panel A and B: please use the same color scheme for both panels, so the STABLE subpanel A should have lighter shades of purple and green as in panel B.

Reply: We apologize for color confusion. The data in panel B are not the same as in A, because they only express the population size of P. veronii, whereas A shows the whole community.

Action: We changed the color profile to make this clearer and specified, additionally, when the samples come from microcosms not exposed to toluene. We also added the sizes of the microcosm communities exposed to toluene with P. veronii added, to better describe this (and reply to request from Reviewer 3).

36) The pie chart and barplot from Figure 3 and Fig. S3 have too many colors and it is not possible to see what is happening. I suggest to use a threshold and only show 10 or 15 taxa above the threshold which can be seen on the plots and which are also described in the text. In addition, provide supplementary tables which list all the taxa which are unregulated or down regulated upon addition of toluene.

Reply and action: In the light of the suggestions to better specify the outlier procedure, we only kept the scatter plots and specify the taxonomy of the outliers by a three-letter code to be conform to Figure 5. All other taxa details are listed in new Tables S1 and S2.

37) Fig. S4: why show colors for all these phyla which are not visible? see comment above. Please only show 10 or 14 taxa, it is not possible with the eyes to see the difference between so many shades.

Reply and action: We apologize again for the color confusion. Figure S4 was removed with the revision of Figure 5, and replaced by a supplementary table S3 with the bead community data.

38) Fig. 4 top of panel. what is the different between the first facet ("mixed C") and the second facet (also "mixed C"). The text at the bottom also seems identical (T0, T48, T48, -, -, Pve).

Reply and action: This referred to independent biological bead experiments – see also Fig. S5. We prefer to keep it, but have added the specification 'Experiment 1', 'Experiment 2')

39) Figure 4: It is confusing to have this abbreviation "SC" in the title, there is enough space to write these words in full. Do not introduce unnecessary abbreviations.

Reply and action: SC was specified to 'soil cells' in the figure names.

40) Figure 5 Panel A: It is not clear to me if a selection of families are shown in this panel or if all the detected families are shown. What are the numbers on the x-axis?

Reply: Based on the other suggestions from this reviewer, we completely revised Figure 5 to show community variation in the bead experiments based on ASV, and to have a comparison with outliers from the microcosm experiments.

41) Figure 5: I don't understand the legend with the different colors at the family level. Why have B Burholderiaceae and C Caulobacteriaceae but no color for those two families? If those

families are absent in the figure, please drop them in the legend. There are some colors in Panel C which are not described in the figure legend, for example orange. Upon reading the legend for the third time, I see that panel C shows ASVs but I find colors are not helpful here and this whole panel is very confusing. Best would be to present these results as supplementary table or drop the panel entirely.

Reply and action: See our remark above. Figure 5 was revised entirely and we have tried to remove as much as possible confusing colors. We now show Panel A: community comparison across bead treatments with and without inoculant (unique ASVs but collapsed to genus level for comparison), B: outlier identification in the beads incubated with either soil extract or toluene as a consequence of paired inoculation with Pve (based on unique ASVs but displayed as a function of Phyla), C: Comparison of outliers in toluene and Pve-inoculated beads and microcosms (collapsed to Family level for the comparison) and D: Comparison of the bead-inoculated communities with Pve and corresponding microcosms by non-metric multidimensional scaling and permanova, to see the general effect of toluene, inoculant and system ('bead' versus 'microcosm').

Reviewer #2 (Remarks to the Author):

This work investigates the importance of nutritional niche availability in allowing introduced bacterial species to thrive within resident soil microbial communities. By comparing introduction of various *Pseudomonads* and *E. coli* to stable versus growing natural communities, the authors find that introduced species proliferate more when co-introduced with a growing soil community and when inoculated into soil amended with an inoculum preferred carbon source like toluene.

The authors also demonstrate through agarose bead encapsulation experiments that the inoculants benefit the growth of resident soil community taxa and that conversely, the presence of different soil community taxa negatively impact growth of inoculant microbes. The authors describe this phenomenon as competitive loss by facilitation whereby inoculant microbe metabolism generates additional metabolites and by-products that soil microbes can utilize. The authors go on to further show that the competitive loss by facilitation is also evident within non-sterile soils that are more representative of in situ conditions. Here they use metatranscriptomics to elucidate abundance of resident communities in response to *P. veronii* inoculation and observe there is an increase in their abundance in the presence of *P. veronii* for some soil types. They also find the resident community has an enrichment for pathways in toluene by-product degradation pathways, indicating a likelihood for cross-feeding between *P. veronii* and the resident microbial community.

1) The authors did a thorough investigation of their hypothesis of competitive loss by facilitation however the novelty of these findings could be more clearly communicated. It is not evident from the way the manuscript is written whether this is a newly described phenomenon or if it is confirming previous studies that have also shown this trend. Given that the manuscript title includes "competitive loss by facilitation", it would be helpful for the authors to discuss whether this phenomenon has been observed previously or whether their study is the first to demonstrate it.

We thank the reviewer for the overall positive summary of our work. Facilitation is a used terminology, for example, in reference to drug resistance formation (<https://www.pnas.org/doi/10.1073/pnas.0707766104>), but has so far not been used in the context of loss of competition to an inoculant.

Action: We revisited lines 133-136 in the introduction, and l. 762.764 in the discussion to explain the concept of the term competitive loss by facilitation.

2) This study uses robust methodology to demonstrate the importance of nutritional niche availability on microbiome engineering, however it does not investigate other factors impacting microbiome engineering like presence of predator microorganisms, environmental tolerance (to abiotic factors in the soil like salinity, pH, etc), effect of inoculum size or dispersal mechanism. Given these other factors are not investigated in this work, the conclusions made should also clearly state they are limited to nutritional niche availability and that other factors may also influence some of the observed results.

Reply and action : We thank the reviewer for these suggestions. Indeed, these factors have been regularly mentioned in older literature as inhibiting the fate of inoculants (see, for example, the introduction between lines 86-88). We mentioned these factors again in l. 875-878.

Comments that require revision before publication:

3) Line 20: the authors should list the ecological factors they study in this work.

Reply and action: We rewrote the abstract – also based on comments from reviewer 1, and hope this is now better explained.

4) Line 24. The way to describe addition of toluene as “inoculant selective niche” seems like an overstatement, especially considering the growing NatCom also appeared to have improved growth in response to toluene. I would recommend rewording it to “inoculant preferred carbon source”.

*Reply : We thank the reviewer for the suggestion, to which we respectfully disagree. Toluene is not a preferred carbon source for *P. veronii* (which also causes toxicity), but it is a nutrient niche that we can deploy to select for its growth. As we discuss in lines 363-368 and 413-416, the nutrient niche is not completely selective, because some background growth may occur. Growth in the soil microcosms with toluene leads to major disturbances (see Fig. S3), and we think it is rather the fact that some taxa are more robust that enables them to profit more from the available carbon sources than that they are truly growing on toluene.*

*Action: We rewrote part of the introduction to explain this more clearly, and also to explain our assumptions on nutrient niche availabilities (l. 126-138). We also added a specific description of the effect on growth of the community by inoculating *P. veronii* to toluene-exposed communities (l. 415-420) and included these results in Figure 3A.*

5) Line 30. The term “selective niche generation” here is not clear. The addition of toluene greatly influenced the diversity of the NatCom (Figure S3) without the presence of any introduced *P. veronii* as well.

Reply and action: We thank the reviewer for this remark. Indeed, toluene itself causes a major disturbance to the community (Figure S3 and l. 413-416, l. 438-440). We refocused here on the perspective of the inoculant – which showed improved survival, and improved growth of the soil community (l. 414-418). In addition, we reanalyzed and compared the outlier taxa data from toluene exposure across the soil microcosm, bead and metatranscriptomic experiments (new Figure 5C).

6) Line 54. Is this accurate there is no previous work that aids in resolving why newly inoculated strains don't establish in target microbiomes? It would be useful to provide some references here to support this otherwise reword "remain unresolved" to "remain largely unresolved".

Reply and action: We have recently made an extensive review of the literature on this subject (cited as ref 16), from which we conclude that there is no good basis to conclude why strains do not survive after inoculation (except in case of pathogens).

Action: Reworded as suggested.

7) Line 56. Is this the correct reference for this sentence? Reference 16 does not describe niche availability as a factor for determining successful inoculant proliferation. Please replace with the correct citation to support this statement.

Reply and action: We apologize. As mentioned above to Reviewer 1, there was a small error in our Endnote referencing. This has been resolved.

8) Line 59-60. Is this the correct reference for this statement? The reference cited describes how transiently invading microbes have a permanent impact on the resident community and does not appear to describe how emerging function of resident microbiome disfavours opportunities for new strains to grow.

Reply and action : Reference was replaced by R. Debray, R. A. Herbert, A. L. Jaffe, A. Crits-Christoph, M. E. Power, B. Koskella, Priority effects in microbiome assembly. Nat Rev Microbiol, (2021).10.1038/s41579-021-00604-w

9) Line 72. This work doesn't discuss or demonstrate through experimentation how the introduced inoculum provides plant-growth beneficial functions. It would be good to see more discussion around this.

Reply and action: We appreciate the point raised but here we only use the P. protegens strain as an example of a bacterium which is frequently mentioned for its plant-beneficial character.

Effects on plants was outside the objectives of the study.

We included a reference to a recent study on P. protegens inoculation into soils and discuss this in line 830-832 of the discussion.

10) Line 77. It would be good to describe the NatCom in more detail here to give context on the abiotic and biotic features of the soil (ie. Presence of predator species? pH, salinity? Available carbon sources?).

Reply: We thank the reviewer for the suggestion. All (and more) NatCom details are provided in the materials and methods section (l. 904-906 and 936-939), and it would become too much to repeat that here in the introduction.

Action: We added here the basic concept of the soil microcosms (l. 125-130).

11) Line 90. When the term “niche availability” is used throughout, it should be more specifically “nutrient niche availability” as this was the only aspect investigated (ie. Through addition of toluene). If the authors also tested different dispersal introduction methods this would examine spatial niche availability and also if they tested different pH gradients this would examine abiotic niche availability.

Reply and action: We thank the reviewer for the suggestion. We specified wherever appropriate that we mean nutrient niches, but in other cases maintain mentioning spatial niches, since a priori we cannot exclude that soil colonization by resident communities would pre-empt available ‘space’ for inoculants.

As example for the more specific use of nutrient niches: the corrected l. 128-135. For a statement on spatial niches, see for example, the new lines 357-359.

12) Line 81 and 91. The term inoculant-specific niche is broad as it seems to be just a nutritional niche that is created. I would recommend changing to “inoculant-specific nutritional niche” or “inoculant preferred carbon source”.

Reply and action: See our action plan to remark 11 to this reviewer.

13) Line 109. It would be helpful to describe the carbon source profile of the soil organic carbon solution and whether the introduced *Pseudomonas* sp. and *E. coli* have the metabolic capacity to utilize the available carbon sources.

Reply: The general nature of the available complex carbon in the soil has been characterized previously, to which we refer in the materials and methods section (l. 850-852). Unfortunately, this is not a list of available carbon compounds because soil organic matter is too complex to determine this.

Action: We change this wording to “a complex organic compound fraction extracted directly from soil to provide nutrients for bacterial growth” (l. 126), to make clear this is not a list of organic compounds that we prepare and add. This was further clarified in the Materials and Methods section l. 904-906.

14) Line 120. Was the transition of the NatCom back to its original state within a month expected? It would be helpful to include a comment on whether this is expected given what is known from previous studies.

Reply: We thank the reviewer for the comment. In a way, yes, from previous studies where we regularly transferred growing soil communities, we expected that they would behave in a somewhat reproducible manner and (much like a pure culture), would go through a (community) growth phase and then some form of a stationary phase. On the other hand, it was the first time we revived a soil community in this manner.

Action: This was rewritten in lines 213-216.

15) Line 125. Were the microcosms split into 45 before phase 1 for a particular reason? Since they were pooled at the end of Phase 1, it is not clear why they were split in the first place.

Reply and action: We thank the reviewer for this observation. In fact, this was just a very practical aspect to maintain consistency of growth with the same volumetric ratios. Otherwise, in order to produce such a big soil community batch, we would have to grow first in a very big container and then split to smaller, which could have affected the outcomes.

We added a statement about this in line 199/l. 255, and in the Materials section, l. 944-945.

16) Line 133. Was it expected the growing NatCom remained lower than stable NatComs during most of phase 2? Is there some level of re-adaptation to the fresh soil microcosm and soil extract that limits their ability to reach as high titres as stable? A sentence or two around whether this was expected would be helpful.

Reply: We thank the reviewer for this question. No, this was not expected. We assume this is an effect of slower growers in the community (given that at the end of Phase 2 the sizes of the STABLE and GROWING were again equal). Unfortunately, there is still relatively little experience with growing complex communities and we don't know exactly which properties cause lower yield.

Action: Rephrased in l. 264-266.

17) Line 148-149. The inoculum size of freshly diluted growing NatCom versus stable NatCom at $t=0$ in phase 2 is different ($\sim 5 \times 10^7$ for stable vs $\sim 5 \times 10^6$ for growing as shown in Fig 1E). This 10-fold difference in inoculum size of NatCom is an additional factor to nutrient replete vs deplete conditions that would affect niche availability for the introduced *Pseudomonas* sp and *E. coli* which were inoculated at a lower inoculum (1×10^5). Hence, the niche availability in growing vs stable is higher both spatially and nutritionally and this should be clearly stated.

Reply : We thank the reviewer for the observation. In fact, this is an inevitable consequence of testing the difference of inoculants into GROWING versus STABLE communities; the latter having higher densities AND depleted the nutrients. It was exactly the state we wanted to create, because only with a STABLE community at higher density at start, the nutrient conditions would have been different. Since we did not analyze the spatial positioning of the various taxa in the soil microcosms, we cannot know whether there is also a spatial niche restriction to the development of the inoculants (in STABLE) versus GROWING. The main argument to assume that there is no spatial constraint on the inoculant's proliferation is the observations that it can still grow to much higher and equivalent levels as the community itself in case of toluene addition (e.g., 1×10^8 cells per gram soil) (Fig. 3B).

Action: We rephrased this in l. 128-134/l. 277-281.

18) Line 173. Since no analysis of the nutrient profile was done to confirm this and also since the growing NatCom condition would also have more spatial niche available, it is more accurate to say the "potential nutrient and spatial niche".

Reply: We thank the reviewer for the comment but respectfully disagree. Bacterial growth is an excellent indicator for the utilization of carbon and other major nutrients. Consequently, if a population develops to 1×10^8 cells per gram of soil (like in the axenic condition); this automatically

means that ca. tenfold more carbon was available for that strain in the system (at a typical yield of 10%). Hence, if the same inoculant in STABLE or GROWING NatCom background only develops to 1e6 cells per gram, they either have less carbon available (our conclusion), or are eaten or killed by something else.

Action: See, for example, the discussion in l. 778-781, and the use of 'realised niche' in l. 350-354, l. 424-426.

19) Also, how is the 1% of nutrient niche available calculated? Is this based on growth of the introduced species? There should be a reference to the methodology for calculating this as it's not obvious how this percentage was obtained.

Reply: We apologize that this was not clear. Essentially our calculation is from the attained population sizes – as explained above in point 13 to Reviewer 1.

Action: This was corrected in l. 352-355 and again in the discussion (l. 753).

20) Line 181. What does a priori predation mean? Was predation measured in this work?

Reply: We referred to an argumentation as (two points) above, but realize this is not clear.

Action: Was removed from text (Reviewer 1 had the same comment)

22) Line 189-190. Was it expected that the growing NatCom would have a higher ability to utilise toluene than the stable NatCom? A sentence here about whether this was expected and why would be helpful.

Reply: We thank the reviewer for the observation. No, this was unexpected. We think it may be due to the large disturbance in growth of most taxa in the NatCom upon toluene addition, which shifted the nutrient utilization profiles.

*Action. We added a sentence along these lines in l. 372, then 413-417. We also add the data on the community growth with *P. veronii* inoculation and toluene exposition, showing that actually *P. veronii* metabolism does lead to higher overall community growth (revised Fig. 3A, and l. 413-417).*

23) Line 195. Has this been reported as well in other studies? A comment about whether this is a novel finding is important to include here.

Reply and action: This has not been demonstrated for the context of a soil microbiome as far as we know, but it has recently been shown for gut microbiota. We refer to this in the discussion in lines 762-764, therefore, we feel it is unnecessary to comment specifically on this here.

24) Line 217-219. The result that toluene had a greater impact on NatCom diversity in the absence of *P. veronii* compared to the presence of *P. veronii* is not explained. This seems important as it shows the effect of toluene greatly shifts NatCom diversity and this effect is dampened by the presence of *P. veronii*. Was this expected? What is the reason this occurs? More discussion around this significant trend is warranted.

*Reply : We thank the reviewer for this observation. Indeed, toluene exposure caused a relatively dramatic change in the growth of many taxa in the NatCom, which was lessened by inoculation of *P. veronii*. We suspect that this is a consequence of replacement of toluene-sensitive taxa by toluene-resistant taxa, and also by the detoxification of toluene by *P. veronii*.*

Action: Also on the basis of Reviewer 1 comments, we revised the detection and description of the outlier effects, and combined to a revised Figure 3E and F (with more specific description of outlier effects as a consequence of toluene addition), the new Figure 5, where we compare the effect in the context of P. veronii inoculation, and revised Figure 3A to show the effect on community size by P. veronii inoculation in presence of toluene. Corresponding new texts l. 413-418 (community growth with toluene and P. veronii), l. 431-435 (outlier procedure and effects), l. 438-502 (toluene effects), and l. 635-645 (comparison of toluene effects among microcosms and beads).

25) Fig S3 A) is missing 2 graphs that it mentions in the caption are included, the without toluene (and no P. veronii) impact on growing and stable NatCom diversity.

Reply and action: As we mention in our reply to remark 24, this was completely re-analyzed and described. See the new Figure S3 for a comparison of the toluene effects.

26) Line 221. Can the authors elaborate on how P. veronii inoculation alleviated negative effects of toluene exposure on NatComs? This is not clearly explained. It is because there were more pronounced taxa changes for NatComs exposure to toluene without P. veronii inoculation? Since Fig S3 is missing two graphs of no toluene or P. veronii exposure, it is not clear if there are also significant taxa changes in this condition.

Reply and action: see our reply to remark 24 in this context. See the new Figure S3 for a comparison of the toluene effects.

27) Line 225. This section needs to be introduced more clearly in terms of how the experiment was designed. There is no rationale given for why sand extract was used and why (and what) 16 carbon sources were used for the Mixed C condition.

Reply: The specifics of the experiment have been extensively described in the materials and methods section, also including the description of the 16-C substrates (l. 1062-1068). We are a little bit afraid that re-explaining all that here in the results section would expand it too much. The main difference here in the design is really the emphasis on forcing (random) but paired interactions of soil cells with the inoculants, which we assumed would lead to stronger magnitude effects than seen in the microcosms. However, since the agarose beads are essentially an in vitro system, it is more complicated to provide the appropriate nutrients for growth of the soil taxa. This is thus why we tested differences between (again) a soil extract (undefined, but as results showed, the best in terms of favoring growth of many different soil taxa), a defined mixture of 16C sources (that had been suggested as a good method to grow soil bacteria, but – we find, less good than soil extract).

Action: We reworded this in l. 513-515 (general idea of the paired bead studies), l. 520-524 (th nutrient conditions in the beads), and l. 594-596/626-633 (comparison of the bead treatments on developing community compositions).

28) Lines 228-234. I found this section difficult to understand without referring to the methods section. It would be helpful for the authors to make it clear that each bead contained on average 1 inoculant cell plus 1 soil NatCom species. Also, clearly state that “founder cell pairs” refers to the ~2 initial cells inoculated into each bead.

Reply: We apologize if this was not clear. See also the cartoons in Fig. 4A.

Action: We rephrased this in line 517-518 and referred to the cartoons in Fig. 4A.

29) Line 239. Do the authors mean “sand” rather than “soil” extract here? According to Fig. 4A, it is sand extract.

Reply: Indeed, but “sand” is also a soil. We stuck to extracting the organic matter fraction from the material where we obtained the soil cells from. As this was also commented on by reviewer 1, we referred to this term in general by “soil extract” as explained in l. 1062-1068.

30) Line 245-246. It would be helpful to have a number stated here for average size of SC Alone vs SC with PV as it is hard to see from Fig. 4A how significant the difference in size is. Also, the authors state that SC alone in mix shows similar average growth to SC alone, however the p-values shown in Fig. 4A suggest there is a significant difference in average growth between SC alone and SC alone in mix.

Reply: We thank the reviewer for this remark. Given the spread in the distribution of microcolony reproductivities (this is a log₁₀-scale), we feel that the mean or median do not give sufficient justice for the comparison. This is the reason we deployed the Fisher’s test to compare the distributions. The reviewer is right that bead distributions with SC alone in the mix are also different from SC alone.

Action: This was removed from l. 568.

31) Line 273. Soil extract is mentioned in text but in Fig. 5 it is described as sand extract. Also same issue in Line 284.

Reply and action: We consolidated to the use of ‘soil extract’ but specified that we use the soil from which the bacteria were isolated to prepare the extract (e.g., l. 1062-1068, and elsewhere).

32) Line 312. What is the possible reason that clay soils did not allow for as much *P. veronii* growth?

Reply: This is an interesting question to which unfortunately we do not have the answer. We suspect this has to do with a higher proportion of bacterial predators but we don’t have sufficient information to support this.

No further action.

33) Were resident microbiota transcripts in clay associated with aromatic compound metabolism and they just outcompeted *P. veronii* for toluene utilization?

Reply and action: This is not supported by the proportion of aromatic compound metabolism-associated transcripts that we find in clay communities compared to the others (Fig. S7); specified in l. 730.

34) Line 316. Did the transcript of silt resident communities become enriched in aromatic compound metabolism later in the incubation? It appears this is the case in Fig 6D. A sentence should be added here to describe this observation as it offers an explanation of why their higher abundance of ribosomal protein transcripts occurred later in incubation as they adapted to toluene utilization. It is also useful for the authors to mention the observation that in the presence of *P. veronii* in silt conditions at later time points, the transcript levels for toluene

degradation pathway genes were lower than that for those in the absence of *P. veronii*. This would suggest the resident soil community adapts more strongly to toluene utilisation pathways in the absence of *P. veronii* which makes it grow slightly less (as seen in Fig. 6B).

Reply and action: Indeed, this is the case and we comment on this in line 723-725. It looks as if the transcript levels for toluene aromatic compound metabolism at the later time point in the microcosm without P. veronii remains higher than in its presence, but this is not statistically significant. Also the growth rates in Fig. 6B are not significantly different for the two microcosm types. For this reason, we did not want to imply anything further here.

35) Line 320. Fig 6D does not show “inoculated toluene-free”. I think the authors meant to say “uninoculated toluene-free” here.

Reply and action: Thank you. Was corrected to ‘uninoculated toluene-free’.

36) Line 324. Fig S6 and S7 do not show the transcript levels for Jonction + toluene but uninoculated with *P. veronii*. This figure would be helpful to include to support the statement that is made in this sentence.

Reply: This is correctly observed, but this is not what is meant here. What we mean is the native transcript levels to aromatic compound metabolism compared to the other types of soils.

Action: We included here ‘in comparison to Silt or Clay’ (l. 727).

37) Line 332. I believe the authors mean Fig. S8 not S7 here.

Reply: Thank you. This is indeed Fig. S8.

38) Line 342. I would recommend rewording to “available nutritional niches” as those are the particular niches that were investigated.

Reply and action: Was corrected and specified throughout the manuscript (see reply to remark 11).

39) Line 347. As mentioned earlier, it would be useful to know how is the 1% calculated?

Reply: This is/was specified in l. 752, but we explain this more clearly in lines 352-355.

40) Line 351-355. It is difficult to understand what the authors are trying to say here. I recommend this sentence is reworded and made clearer. Also, given that the initial inoculum size of the growing community is higher than that of the *Pseudomonas* inoculants, the fact that more spatial niche will be occupied initially by the growing community is also worth mentioning.

Reply and action: See again our reply to the general remark on niche utilization (point 11). We return to this in the discussion in l. 822-825.

41) Line 356. Not sure if the term “leakage of metabolites” is accurate here, consider rewording or explain what this means more clearly. For instance, does it refer to the fact that some of the resident soil community is able to utilise the toluene on its own and thus they are able to use

this as a carbon source in addition to the by-products of toluene degradation generated by *Pseudomonas* toluene metabolism?

Reply and action: this is explained and discussed in more detail in l. 763-770.

42) Line 361. The concept of growth inhibition of the inoculant warrants further explanation given it is mentioned for the first time in the manuscript here. How does it differ from competitive loss? Was there any investigation into growth inhibiting factors?

Reply: We thank the reviewer for this comment. However, we mention growth inhibitors in the discussion in line 356 already, as a possibility to explain why inoculants in bead pairs with soil cells proliferate less well than alone. Currently it is not possible to measure in single agarose bead pairs whether some strain would excrete a potential inhibitor; therefore, we can only compare productivity distributions. At least for toluene metabolism we could measure the induction of related pathways from toluene metabolites in resident bacteria, which we use as argument to state that facilitation and competitive loss is more common than pure inhibition. Action: we rephrased this more specifically in l. 768-770.

43) Line 368-369. Is this stated expectation based on previous literature or a hypothesis? Include citations if based on previous literature. Also, the conclusions made in line 374-375 do not correspond with the stated expectation that growing communities would limit proliferation of incoming species since in line 373, the authors state the observed result that the inoculants established better in growing than stable NatCom conditions. This section needs to be re-worded to improve clarity.

Reply: We appreciate the suggestion. We think that substrate-limited growth is a general well-supported idea in microbial ecology, cited in reference 21. We did not wish to imply that GROWING communities would limit proliferation more; rather, we tested and demonstrated that in presence of GROWING communities (in soil), there are still more nutrients available to an inoculant. Action: This was rephrased in lines 779-781 and then 823-825.

44) Line 378. The authors should be more specific they are observing niche “nutrient” (un)availability and competition “for nutrients”.

Reply and action: See you reply to the general remark 11 by the reviewer.

45) Line 382. Is this the first study to show this is applicable to soil microbiota interventions? If no then include citations of other studies that show this. If yes, then state it explicitly as it highlights the impact of this work.

Reply: We appreciate this suggestion. Yes, we believe this is the first time we show the effect of nutrient niches directly in the context of soil microbiomes. See line 832 and l. 145-149.

46) Line 385. “potential biological interactions” sounds too vague. I would suggest re-wording this to “potential interspecies cross-feeding interactions”.

*Reply: We thank the reviewer for the suggestion, but a priori, when pairing soil taxa cells with inoculants inside agarose beads, we cannot know which type of interactions might prevail. Therefore, we prefer to stick with 'potential biological interactions'.
Action: we specified here 'such as substrate competition or metabolite cross-feeding' (l. 835).*

47) Line 390. "facilitating the growth of other soil bacteria" is also too vague. I would recommend the authors be more specific what mode of facilitation they identified in their work (ie. Cross-feeding).

Reply and action: We appreciate the suggestion. We rephrased paragraph in l. 763-770 on facilitation.

48) Line 400-402. The concept of temporal available niches is brought up for the first time here and should be defined. It's not clear how the results of this study demonstrates establishment of temporal niches. This discussion point seems out of place and should be clarified.

Reply: We have clearly raised this point as an outlook on the potential to establish niches for engineering. We did not claim that our results demonstrate establishment of temporal niches. No further action.

49) Line 405. I would specify "nutritional niche engineering" as this study mainly focused on investigating nutritional niches.

Reply and action: See reply to remark 11.

50) Line 409. The concluding sentences need some additional work. For instance, this study showed that bioremediation agents would not exactly "thrive thanks to the selective niche provided" but instead be restricted through competitive loss by facilitation. I think the authors need a stronger conclusion on how their work can inform microbiome engineering for bioremediation purposes.

*Reply : We thank the reviewer for the support but we respectfully disagree to the interpretation. We show very clearly how addition of toluene is beneficial for P. veronii as an inoculant (without which it would only establish at low levels in the community – see Fig. 3). Without this, the inoculant cannot 'thrive' at all. Simultaneously, however, the inoculant will lose part of its potential productivity by engaging in metabolite sharing. Therefore, our last concluding sentence is warranted.
Action: no further changes, but the last sentence of the abstract was reworked to specify this better.*

51) Line 441. Can the authors describe the expected carbon sources present in the forest soil extract? This would help give context as to why E. coli had poor growth in the soil microcosms.

*Reply : Unfortunately we don't have a precise list of compounds present in forest soil extract, but could only globally characterize the nature of its organic matter (see Ref. 30). This is not uncommon, since soil organic matter is a very complex mixture of compounds. Our reply to the specification of organic matter is presented at remark 13.
No further action.*

52) Fig 1 A). make bottle grey to not confuse with the different shades of blue to describe NatCom in soil (18 mo and 1 mo). Also make fresh soil + extract vs 1 mo old soil + extract Phase 1.

Reply and action: Colors were changed here; suggestion incorporated.

53) Fig S3 B), Fig S6 and Fig S8 - the font in these figures is too small.

Reply: We thank the reviewer for the suggestion. Fig. S3B was corrected, but it is impossible to make fonts bigger for Fig. S6 and S8 (and not lose the link to the table). Since these are electronic (supplementary) PDFs, they can easily be enlarged on the screen for visibility.

Action: no further action.

Reviewer #3 (Remarks to the Author):

Review of "Niche Availability and Competitive Loss by Facilitation Control Proliferation of Bacterial Strains Intended for Soil Microbiome Interventions" by Čaušević et al.

****Manuscript ID:**** NCOMMS-23-49794

In the paper titled "Niche Availability and Competitive Loss by Facilitation Control Proliferation of Bacterial Strains Intended for Soil Microbiome Interventions," Senka Čaušević et al. explore the complex dynamics of introducing bacterial inoculants into soil communities. The study primarily focuses on understanding the proliferation of typical soil inoculants, such as *Pseudomonas* spp., in relation to niche availability and competition within diverse soil microbial communities (NatComs). The research employs a combination of methodologies, including the cultivation of standardized soil microbiomes, randomized paired growth assays, and metatranscriptomic analysis. The findings reveal that the proliferation of soil inoculants is influenced by limited niche availability and their competitive disadvantage arising from facilitating the growth of rival populations. This situation can be altered by creating selective niches, such as introducing bioavailable toluene, which enhances inoculant survival and integration into the soil microbiome. This work underscores the significance of niche dynamics in microbial community interventions and suggests that strategic niche manipulation can improve the successful integration of inoculants into soil ecosystems. The results offer crucial insights for microbiome engineering, particularly in the realms of soil health and bioremediation strategies.

We thank the reviewer for the overall positive summary of our work.

1) While the paper is well-articulated and offers a novel perspective on the interactions of bacterial inoculants within soil microbiomes, it would greatly benefit from a deeper re-analysis of the data to provide a more comprehensive understanding of the results. This includes a more detailed and nuanced description of how the inoculants affect community composition, as well as a robust comparative analysis of the inoculant's impact on the entire community versus its effect in random pairwise experiments. Such an analysis could reveal intricate patterns and dynamics that are not immediately apparent, thereby enriching the study's conclusions.

Addressing these aspects is crucial for a more complete and accurate representation of the microbial interactions and their ecological implications. Thus, I recommend publication, provided that the authors undertake this comprehensive data re-analysis and enhance the discussion to more effectively contextualize and interpret the study's findings within the broader scope of microbial ecology and soil microbiome research.

We thank the reviewer for these suggestions. In line with reviewer 1 suggestions, we have revisited the statistical procedures to detect and conclude potential shifts in community composition as an effect of inoculant addition. Based on reviewer 2 comments, we have reanalyzed the effects of toluene addition across communities.

Action plan: We have revised Figure 3E&F, and Figure 5 and included a comparison between bead communities and microcosm communities to detect commonly enriched or depleted taxa. The corresponding paragraphs in the results section were rewritten. See our reply to remark 24 and 27 of reviewer 2.

Comments on the Abstract

2) The last sentence of the Abstract contains the phrase "competitive loss by facilitation," which may lead to ambiguity. In ecological contexts, "facilitation" typically denotes a positive interaction benefiting at least one participant, usually without notable detriment to the facilitator. Therefore, the expression "competitive loss by facilitation" could be misinterpreted as it implies a reduction in competitive ability through a mechanism generally viewed as advantageous.

Reply: We thank the reviewer for the suggestion. Based on the recommendations of reviewers 1 and 2, we think that the terminology 'competitive loss by facilitation' covers what we conclude is happening here, and is also the main new finding. But formally, the ambiguity is also correct, because the facilitation benefits one participant, and competitive loss reduces the ability of the inoculant to proliferate.

Action : The abstract was rewritten based on suggestions by reviewer 1.

Comments on the Introduction:

1. In the introduction, lines 56 to 63 explore reasons why inoculant bacteria might struggle to establish themselves in soil microbial communities, focusing on competition for resources and habitat changes that impede their survival or interaction with resident microbes. The subsequent lines, 63 to 65, point out a gap: these theories, particularly regarding resource competition and microbial interactions, haven't been rigorously tested in experiments involving both the microbiome and inoculants. Moreover, the text notes that existing mechanistic theories are largely based on a narrow range of strains, often pathogenic. This raises a critical question: What exactly are these "mechanistic concepts," and how do they connect to the ecological processes discussed earlier? Are these concepts applicable to the broader dynamics of soil microbial communities, or are they too limited in scope?

Reply: We thank the reviewer for the conceptual remarks. With ‘mechanistic concepts’, we mean such things as virulence factors that can aid pathogenic bacteria to induce or recover a niche within a system.

Action: This was changed to “..mechanistic understanding of inoculant proliferation is based on (...) studies with frequently, invasive pathogenic strains” (l. 90-91).

2. Similarly, the reference to "underlying mechanisms" in lines 65-68 remains vague. The text mentions "N+1/N–1 engineering," a concept purportedly useful for revealing mechanisms of community assembly and development, yet fails to elucidate what this concept entails. This omission leaves a gap in the introduction's conceptual framework, raising more questions than it answers. Additionally, the current wording does not clarify whether N+1/N–1 engineering applies solely to pathogenic strains or has broader applicability. For an introduction to effectively set the stage for the results, it should clearly articulate these underlying mechanisms, especially those influencing the success or failure of inoculant strains in various ecological contexts. Simply stating the existence of a mechanism or limiting discussion to pathogenic strains, without adequately explaining the introduced concept, falls short of providing the necessary context for the study's findings.

Reply: We appreciate the reviewer’s remarks here. We refer to a longer review that recently appeared on N+1/N–1 engineering (Ref. 16) and which we feel we do not have to repeat here. This review is not based solely on pathogenic bacteria but rather as we state in l. 92 is more general on “how selective inoculation (...) into defined microbiota (...) can be used to uncover underlying mechanisms”. We believe this is sufficiently clear here.

Action plan: We specified better than this does not only concern pathogenic strains (l. 90-91), and further refocused the introduction to the study design here (l. 126-138).

General Comment for the Results section:

3) In reviewing the manuscript, I suggest that the authors carefully reconsider the placement of certain details within the Results section. Specifically, information such as, "All inoculants constitutively expressed an introduced genetically encoded mCherry tag, which facilitated the quantification of their specific population size," appears more appropriate for the Materials and Methods section. The inclusion of such methodological details within the Results section disrupts the narrative flow and impedes a clear and concise presentation of the findings. Relocating these details would enhance the readability and coherence of the Results section.

Reply: We thank the reviewer for the suggestion, which, however, is not commonly shared among all reviewers. For example, Reviewer 2 would like to see more specific details of methods within the results.

Action: We have tried to find a fair balance and have included different connecting sentences to improve the logic flow of the results sections. This particular mention of the fluorescent tagging was removed.

4) Additionally, it is imperative that the authors refine the logical progression of the Results section. The current structure presents the findings as a series of isolated observations rather than a cohesive narrative. A more effective approach would be to ensure that each segment within the Results logically connects to the next. This will facilitate a comprehensive understanding of the research, allowing readers to follow the progression of the study and appreciate how each result builds upon the previous ones, ultimately contributing to a unified

understanding of the underlying processes. Such a structured approach is crucial for effectively communicating the significance and implications of the research findings.

Reply and action: It was not our intention to present isolated observations and we did make a narrative which in our mind was logic. We apologize that we have not been able to sufficiently guide this narrative here to the reviewer, and have done our best to improve the logic connections between paragraphs – including suggestions from the other reviewers. For example, the new lines 273-274, then 277-282, etc (for each results section).

5. In Lines 136-138, the authors state, “taxa richness was initially lower than that of STABLE NatComs, but became similar from Day 14 onwards (Fig. 1F). This apparent lower diversity...” In this context, there appears to be a conflation between ‘taxa richness’ and ‘diversity.’ Taxa richness pertains solely to the number of species present, whereas diversity is a more comprehensive metric that includes both the number of species and their relative population sizes. The distinction between these two concepts is critical in ecological studies, yet the authors seem to use them interchangeably.

Reply and action : We thank the reviewer for the recommendation to be strict(er) in terminology. We know the difference in the terms (and in the calculations); and have done our very best to comb through the revised text to ensure we use the right term at the right place.

6) The manuscript predominantly features taxa richness (as depicted in Fig. 1F) while relegating the analysis of diversity to the supplemental material (Fig. S1). This decision is not explicitly rationalized in the text. The criticality of this choice becomes evident in the concluding sentence of this section: “We thus concluded that, while the dynamic succession of GROWING and STABLE NatComs varied, the equivalent richness and size of either community meant they were suitable for testing the effect on inoculant proliferation.” This conclusion implies that having equivalent taxa richness is a necessary condition for the GROWING and STABLE NatComs to be considered appropriate for the inoculant proliferation experiments. However, this inference potentially overlooks the importance of diversity in determining the ecological dynamics and experimental outcomes.

Reply : It was not our intention to minimize the importance of any parameter here. The importance of the experimental tests here was to reasonably ensure that we could compare two states of the soil community, which we expected would give rise to differences in available nutrient niches for the inoculants (GROWING and STABLE). For this, we needed to have a comparable idea of both the community compositions and their sizes. Given that we compare a growing system, we need to take into consideration the changes in population sizes of the different taxa, as well as the complete community size. Taxa richness is thus an important criterium, because it sets the stage for the onset of growth.

Action: We rewrote this paragraph (l. 211-216), while revisiting the statistics (based on comments from Reviewer 1). Rather than showing another richness plot (Fig. 1F), we added an ordination analysis of GROWING and STABLE NatComs in Phase 2 to better indicate their global similarity (new Fig. 1F, l. 269-274).

7) Furthermore, there is an issue with the clarity and precision in the reporting of statistical results. On Day 56 in Fig. S1, the p-value is ambiguously reported as “p=00618,” leaving it unclear whether this should be interpreted as p=0.00618 or p=0.0618. This distinction is crucial,

as it determines the statistical significance of the diversity measures. In a scientific context where precise and unambiguous reporting of statistical data is paramount, this lack of clarity could lead to misinterpretations, casting doubt on the validity of the study's conclusions.

Reply: We apologize for this typographic error, which was corrected. To our defense, all statistical test results were already part of the S1 dataset, which was provided.

8) In summary, the authors' treatment of taxa richness and diversity raises concerns regarding both the ecological interpretation and the statistical robustness of their findings.

Reply : With all due respect, we have comprehensively reported all diversity measures on the systems in question – see also the S1 dataset with the specificities of all statistical tests.

Action: See our reply and changes to remarks 5-7.

8). In addition to the previously discussed issues regarding the interpretation and reporting of taxa richness and diversity, a further exploration into the study's approach to community composition metrics is necessary. The authors have presented measures of abundances (FCM counts per gram of soil) and richness (Chao1 index value) in Fig. 1E and Fig. 1F, as well as diversity (Shannon index value in Fig. S1). These metrics, while informative, may not fully capture the complexities of community dynamics.

A key question arises in this context: Can one infer whether a sample belongs to the GROWING or STABLE scenario based on its specific relative abundance profile? Addressing this question would significantly enhance our understanding of the community dynamics at play. The current methodology, focusing primarily on simpler metrics like richness and abundance, may overlook the intricate patterns that emerge from more detailed analyses of relative abundance profiles.

To address this gap, it would be advantageous for the authors to consider applying advanced analytical methods, such as K-means clustering analysis. Such an approach could elucidate whether distinct relative abundance profiles are characteristic of the GROWING and STABLE community states, and whether these states form separate clusters. This analysis would not only provide a deeper insight into the community structures but also add robustness to the conclusions drawn about the dynamics and physiological states of the communities under study.

Reply: We thank the reviewer for these suggestions. Indeed, the community dynamics and the various treatments pose a non-trivial task on the comparison methods. To our defense, we did not only rely on richness, growth and diversity measures, but also on principal component analysis and non-metric multidimensional scale analysis.

Action : We implemented further tools to consolidate the interpretation of community changes, also upon recommendations of reviewers 1 and 2. Notably, we included a new non-metric multidimensional scale analysis, combined with PERMANOVA to study underlying determining factors (New Fig. 1F). We also revisited the analysis of the effects on inoculants by linear regression analysis, and compared microcosm exposure effects to toluene with that of the paired inoculant bead communities (new Figure 5). Immediate text rephrasing here: l. 269-274.

9). Lines 210-211: "In contrast, more dramatic shifts in taxa abundances were observed in presence of toluene and, perhaps counterintuitively, more in STABLE than GROWING NatCom

(Fig. 3D).” Could the authors elucidate the reasons behind these significant shifts in taxa abundances?

Reply and action: We thank the reviewer for the suggestion. In agreement with our reply to Reviewer 2 at point 4-6, we think this comes from changes in toluene-sensitive and -resistant taxa. See l. 372, 413-418. We also specified more clearly the increase in community size that is caused by P. veronii inoculation in toluene-exposed microcosms (Fig. 3A).

10. The initial sections of the results focus on the impact of the inoculum on complex soil communities, while the fourth section adopts a more reductionist yet quantitatively precise approach, examining effects through random pairwise trials. This distinction is noteworthy and praiseworthy for its comprehensive analysis. An intriguing question arises from this: can we link the growth patterns observed in these trials to those seen when inoculants interact with the full community? Specifically, is there a way to correlate species' growth rankings from pairwise trials with their behavior in the context of the entire community? Such analysis would be valuable in quantifying the extent and intensity of nonlinear interactions that occur in a diverse community setting. This comparison would enhance our understanding of how inoculants interact differently in isolated pairwise setups versus more complex, multi-member community environments.

Reply: We thank the reviewer for this important suggestion. We have tried to accomplish this in two ways (see also the reply to reviewer 1 in point 4-6): focusing on the outlier taxa appearing in both toluene-exposed systems (soil microcosms and beads), and, secondly, a general comparison of the communities (but necessarily collapsed to the level of genera, because the ASVs directly are too different among the soils).

Action : This is now reported in a revised Figure 3C-F, with the outlier definition and general differences in the appearance of outliers (l. 431-434, 438-448 and 496-503), and the results conclusion in l. 505-510; and the new Figure 5 (l. 635-644).

Comments on the Discussion section

11) The discussion section tends to revisit the results in excessive detail, which could be streamlined for brevity. A more effective approach would be to succinctly recap the key findings and then focus on integrating these results within the context of existing literature. Each time the results are referenced, they should be directly connected to relevant studies, thus situating the paper's contributions within the broader scientific discourse. This approach would not only enhance the clarity of the discussion but also underscore the significance and novelty of the study's findings in advancing our understanding of the field.

Reply and action: We thank the reviewer for the suggestion. We have revisited the discussion to streamline and recap as much as possible in the first part (l. 762-770), while taking into consideration the recommendations of the other two reviewers.

REVIEWERS' COMMENTS

Reviewer #1 (Remarks to the Author):

I read the rebuttal letter carefully and I am satisfied with the answers from the authors. I appreciate the work they did to improve the writing as well as the figures and I feel the new version of the manuscript is much improved and ready for publication.

Reviewer #2 (Remarks to the Author):

The authors have sufficiently addressed my comments and I accept the manuscript for publication.

Reviewer #3 (Remarks to the Author):

Having considered the authors' responses to the initial concerns—which were also noted by fellow reviewers — I now have no remaining questions or concerns about the manuscript.